# ATLAS: LEARNING TO OPTIMALLY MEMORIZE THE CONTEXT AT TEST TIME

## ABSTRACT

Transformers have been established as the most popular backbones in sequence modeling, mainly due to their effectiveness in in-context retrieval tasks and the ability to learn at scale. Their quadratic memory and time complexity, however, bound their applicability in longer sequences and so has motivated researchers to explore effective alternative architectures such as modern recurrent neural networks (a.k.a long-term recurrent memory module). Despite their recent success in diverse downstream tasks, they struggle in tasks that requires long context understanding and extrapolation to longer sequences. We observe that these shortcomings come from three disjoint aspects in their design: (1) limited memory capacity that is bounded by the architecture of memory and feature mapping of the input; (2) online nature of update, i.e., optimizing the memory only with respect to the last input; and (3) less expressive management of their fixed-size memory. To enhance all these three aspects, we present ATLAS, a long-term memory module with high capacity that learns to memorize the context by optimizing the memory based on the current and past tokens, overcoming the online nature of long-term memory models. Our experimental results on language modeling, common-sense reasoning, recall-intensive, and long-context understanding tasks support the effectiveness of ATLAS compared to other modern recurrent neural networks.

## 1 INTRODUCTION

The attention module (Bahdanau et al., 2014) is a critical building block in modern deep architectures (Vaswani et al., 2017; Achiam et al., 2023; Behrouz et al., 2024; Kamath et al., 2025), excelling due to its scalability and performance in in-context retrieval tasks. In principle, attention functions as an associative memory, computing direct pairwise token dependencies to store key-value mappings and retrieve them via query-key similarities. Computing this pairwise dependencies, however, while accurate, causes quadratic space and time complexity, limiting their applicability in long context understanding, memorization, or modeling (Liu et al., 2024b; Li et al., 2024a; Dalal et al., 2025).

Recent research efforts aim to overcome the limitations of Transformers—i.e., pure attention-based architectures—in long-context modeling by designing more efficient yet effective recurrent neural networks (Schlag et al., 2021; Behrouz et al., 2024; Peng et al., 2025). These modern recurrent architectures can be unified as associative memory modules optimizing an internal objective termed 'attentional bias' (Behrouz et al., 2025). Unlike Transformers' growing KV cache, these models use fixed-size memory, necessitating improved memory management. Consequently, there's growing interest in enhancing RNN memory management through more effective: (i) Learning rules, from additive learning (Katharopoulos et al., 2020) to DeltaNet's Delta rule (Schlag et al., 2021); (ii) Forget (Retention) Gates, from RetNet's input-independent gating (Sun et al., 2023) to adaptive gating in Titans (Behrouz et al., 2024) and RWKV-7 (Peng et al., 2025); and (iii) Memory Architectures, from vector-valued memory (Sun et al., 2023; Peng et al., 2023) to neural deep memory modules (Behrouz et al., 2024; Sun et al., 2024).

Despite the success of these improved models in a diverse set of downstream benchmarks, they often struggle with long context understanding, in-context retrieval, and extrapolation to longer sequences (Wen et al., 2024; Behrouz et al., 2024; Arora et al., 2024; Yang et al., 2024a). We observe these shortcomings arise from three design aspects: (1) The online nature of their memory update, where memory is optimized based on the current token while retaining past memory state, leading

to memorization of individual tokens without considering broader context; (2) The limited capacity of memory, where architecture and key-value feature mappings restrict the number of perfectly mappable key-value pairs; and (3) The expressiveness of memory management (i.e., the internal objective's optimizer), as most recent models use gradient descent that relies on the first-order information about the dynamics of tokens, causing the memory to converge to spurious local minima and learn less effective key-value mappings.

In this paper, we aim to improve the abovementioned limitations—i.e., (1) online nature, (2) limited memory capacity, and (3) less expressive memory management—by designing a long-term neural memory module with high capacity and the ability to memorize the context, instead of tokens. More specifically:

**Better Understanding of Memory Capacity and its Bottleneck.** To improve the limited memory capacity, we suggest using higher-order feature mappings (e.g., polynomial feature kernels) on input tokens. While such kernels often have been used to approximate the sortmax attention, we provide a new theoretical justifications on why deeper memory modules and/or higher-order feature mapping can enhance memory capacity—i.e., the maximum number of linearly independent key-value associations the memory can perfectly map.

**New Expressive Learning Rule.** To improve the online nature of recent recurrent models, this work presents a sliding window update rule, called Omega rule, that optimizes and updates memory based on all past tokens in a given context window, not just the last. This allows the model to better manage its fixed-size memory and memorize a local context instead of individual tokens.

**New Memory Modules with Better Memory Management.** Building upon the above improvements, we present OMEGANET, a new architecture using polynomial features on its keys and queries, while updating its memory based on Omega rule and gradient descent. To further enhance memory management, we introduce ATLAS, which leverages the Muon optimizer (Jordan et al., 2024) as the inner optimization process of the internal memory. We show that both OMEGANET and ATLAS can take advantage of parallelizable training algorithms, resulting in fast training without substantial overhead compared to the online version (i.e., context window = 1). To the best of our knowledge, ATLAS is the first parallelizable recurrent architecture that optimizes the memory using the (approximation) of second-order information (i.e., has locally optimal memory module).

**Improvement on Diverse Downstream Tasks.** Extensive experiments support our model designs: We evaluate OMEGANET, and ATLAS on diverse benchmarks—language modeling, common-sense reasoning, recall-intensive, and needle-in-haystack tasks—where they achieve higher accuracy compared to modern linear RNNs and memory bounded local attention. Furthermore, we studied the effects of memory architecture, feature mapping, memory management algorithm (internal optimizer), and Omega rule on memory module capacity and performance in long-context understanding tasks.

Proofs, additional experimental results, discussions on related work, and the details of experiments are in Appendix.

## 2 PRELIMINARIES

In this section, we first discuss the notation that we use through the paper and then review the background concepts and related work. Additional discussion on related studies are in Appendix B. The extended discussion on background concepts such as attention and linear RNNs are in Appendix C.

**Notations.** We let $x \in \mathbb{R}^{N \times d_{\text{in}}}$ be the input, $\mathcal{M}_t$ be the state of memory $\mathcal{M}$ at time $t$, $\mathbf{K}$ be the keys, $\mathbf{V}$ be the values, and $\mathbf{Q}$ be the query matrices. We use bold lowercase letters with subscript $t$ to refer to vectors correspond to time $t$ (i.e., $\boldsymbol{k}_t$, $\mathbf{v}_t$, and $\boldsymbol{q}_t$). Following Behrouz et al. (2025), we use $\ell(\mathcal{M}_t; \boldsymbol{k}_t, \mathbf{v}_t)$ to refer to the attentional bias (i.e., the internal memory objective). Through the paper, we use simple MLPs with $\mathcal{L}_{\mathcal{M}} \geq 1$ layers and residual connection as the architecture of the memory module $\mathcal{M}(\cdot)$. Notably, despite this choice, all of our model formulations are simply adaptable to other memory architecture choices; e.g., linear matrix-valued memory ($\mathcal{L}_{\mathcal{M}} = 1$). When it is needed, we parameterized the memory module with $\boldsymbol{\theta}_{\mathcal{M}} := \{W_1, \dots, W_{\mathcal{L}_{\mathcal{M}}}, \dots\}$, which at least includes the parameters of linear layers in the MLP.

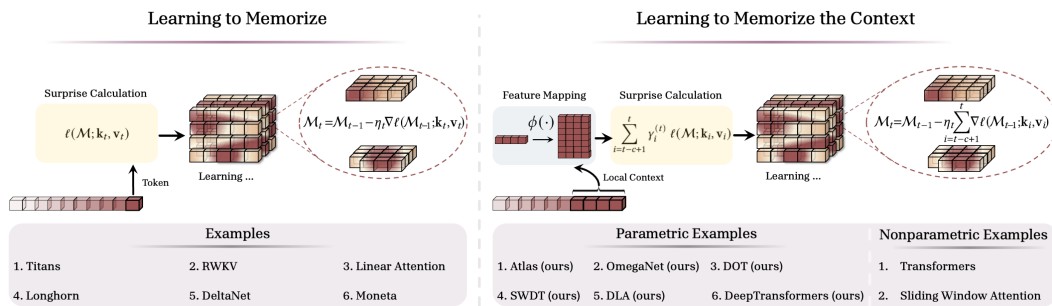

Figure 1: Comparison of learning to memorize (**Left**) individual tokens, and (**Right**) the context.

**Deep Memory Module.** To overcome the limited expressivity of memory and to enhance the *effective* context length of recurrent models, recent studies focus on a new line of architectures with deep memory modules (Irie et al., 2021; Sun et al., 2024; Behrouz et al., 2024; 2025; Wang et al., 2025). These architectures are built on the meta-learning perspective, where the memory is a deep MLP architecture updated by gradient descent (with momentum). Recently, Behrouz et al. (2025) present a framework to accurately unifies popular sequence models as the instances of test time memorization. That is, sequence models are associative memory modules that aim to learn the underlying mapping between given keys and values by optimizing an internal memory objective, called attentional bias. This optimization is based on an iterative optimization algorithms such as gradient descent. More formally, associative memory is defined as:

**Definition 1** (Behrouz et al. (2025)). *Given a set of keys $\mathcal{K} \subseteq \mathbb{R}^{d_k}$ and values $\mathcal{V} \subseteq \mathbb{R}^{d_v}$, associative memory is an mapping $\mathcal{M} : \mathcal{K} \to \mathcal{V}$. Learning the associative memory is based on an objective $\mathcal{L}$, called* Attentional Bias*, that determines the type of memory and its priorities:*

$$\mathcal{M}^* = \arg\min_{\mathcal{M}} \quad \mathcal{L}(\mathcal{M}(\mathcal{K}); \mathcal{V}). \tag{1}$$

Optimizing this objective using an iterative algorithm (e.g., gradient descent) results in the memory update rule. Thus, the sequence model is a meta in-context learner with two optimization levels:

1. Inner Loop: Where parameters of the memory module are optimized (i.e., $\boldsymbol{\theta}_{\mathcal{M}} = \{W_1, W_2, \ldots, W_{\mathcal{L}_{\mathcal{M}},\ldots}\}$). In the inner optimization loop, all other parameters from the model are considered hyperparameters and are fixed and *not* optimized.

2. Outer Loop: Where all other parameters of the model are optimized, such as linear projections, MLP layers, convolutions, etc.

Our terminology builds on this framework. Therefore, instead of full recurrent formulations, we describe models by their: (1) memory architecture, (2) internal objective (i.e., attentional bias), and (3) memory learning algorithm (optimizer). In most cases, models use matrix-valued memory with online gradient descent; for brevity in such instances, we refer to an architecture solely by its internal memory objective. For additional discussions and examples, see Appendix D.

## 3 LEARNING TO MEMORIZE THE CONTEXT AT TEST TIME

Long-term associative memory, crucial for human learning (Terry, 2017), has inspired many artificial neural architectures (He et al., 2024; Krotov & Hopfield, 2016; Schmidhuber & Hochreiter, 1997; Ramsauer et al., 2021; Hopfield, 1982; Behrouz et al., 2024; 2025). While many such models use matrix- or vector-valued memory to compress past data (Von Oswald et al., 2023; Yang et al., 2024a; Schlag et al., 2021), recent studies advocate for deep non-linear neural memory that encodes past abstractions into its parameters (Sun et al., 2024; Behrouz et al., 2024; 2025; Dalal et al., 2025). For long-context reasoning/understanding, however, these long-term neural memory modules still require: (1) High capacity—the maximum (key, value) pairs storable in parameters (see §3.1); (2) A powerful internal memory objective (i.e., *attentional bias*) to learn complex mapping between keys and values (see §3.2); (3) Powerful memory management for better fixed-size memory management (see §3.2); and (4) An efficient parallel training process for large-scale training on modern accelerators (see §H).

This section further discusses these challenges and presents Omega rule: an expressive memory update rule with direct access to tokens in a local context window, which memorizes context rather than individual tokens.

### 3.1 Associative Memory with Super Linear Capacity

As previously discussed, an effective long-term memory module should store past data abstractions in its parameters. However, with a fixed number of memory parameters, a key unanswered question remains: *"what is the maximum number of uncorrelated (key, value) pairs that a model can store?"* To answer this, we start with the simplest case: matrix memory, an $\ell_2$ regression loss as the attentional bias (i.e., $\ell(\mathcal{M}_t; \boldsymbol{k}_t, \mathbf{v}_t) = \|\mathcal{M}_t(\boldsymbol{k}_t) - \mathbf{v}_t\|_2^2$), optimized by gradient descent:

**Proposition 1** (Capacity of $\ell_2$ Attentional Bias). *Let $\mathcal{M}$ be a matrix-valued memory with $d_v \times d_k$ parameters that optimizes the internal objective of $\ell(\mathcal{M}_t; \boldsymbol{k}_t, \mathbf{v}_t) = \|\mathcal{M}_t \boldsymbol{k}_t - \mathbf{v}_t\|_2^2$ with gradient descent. $\mathcal{M}$ can store the mapping of at most $\mathcal{O}(d_k)$ pairs of $(\boldsymbol{k}_i, \mathbf{v}_i)$ with linearly independent keys.*

The above proposition indicates that matrix-valued memory with delta update rule has sub-linear capacity with respect to its number of parameters. This means that the number of independent patterns that can be stored in a fixed-size memory with size $M$ is strictly less than $c \times M$, for some $c \in \mathbb{R}^+$. Recent recurrent models suggest using deep memory modules to store the abstraction of the past into the parameters of a deep neural network (Irie et al., 2021; Sun et al., 2024; Behrouz et al., 2024; 2025). While these deep memory architectures can intuitively enhance the expressive power in modeling complex underlying mapping patterns between keys and values, it is still unclear that if they enhance the memory capacity.

**Theorem 1** (Effect of Deep Memory). *Let $\mathcal{M}(\cdot)$ be an MLP with $\mathcal{L}_\mathcal{M} \geq 2$ layers, $d_k$ input dimension, and $d_h$ hidden dimension. Then, $\mathcal{M}(\cdot)$ can store the mapping of at least $\mathcal{O}(d_k d_v)$ and at most $\mathcal{O}\left(d_k d_v \sum_{i=1}^{\mathcal{L}_\mathcal{M}} \min\{d_h^{(j)}\}_{j \geq i} d_h^{(j+1)}\right)$ pairs of $(\boldsymbol{k}_i, \mathbf{v}_i)$ with linearly independent keys.*

This theorem indicates that deep memory not only improves representational power but also further boosts network capacity, with advantages growing with depth. However, the upper bound remains subquadratic in key and value dimensions, raising the question if a long-term memory module can achieve super-linear capacity.

As stated earlier, the dimension of $\boldsymbol{k}_t$s is crucial for increasing memory capacity. Simply increasing all key and value dimensions, however, significantly increase the number of parameters ($\mathcal{O}(d_{\text{in}})$ per each extra dimension) and memory usage, particularly with long contexts. To address this, building on methods from Kacham et al. (2024a); Krotov & Hopfield (2016), we suggest using separable kernels $\sigma(x, y) = \phi(x)^\top \phi(y)$ for keys and queries. As an example of such kernels, we focus on polynomial kernels of degree at most $p$ to increase input dimensionality and thus network capacity. Given $p \in \mathbb{N}$, let $\phi_p(x) = [x^\beta]_{|\beta| \leq p}$ be a polynomial mapping of $x$ with degree at most $p$. We redefine the associative memory module in Definition 1 by replacing the inner objective of $\mathcal{L}(\mathcal{M}(\mathcal{K}); \mathcal{V})$ with $\mathcal{L}(\mathcal{M}(\phi(\mathcal{K})); \mathcal{V})$. This polynomial mapping enhances representational power by increasing the effective dimensionality of keys without additional parameter overhead for the input projections. Next, we discuss their effect on memory capacity:

**Proposition 2** (Memory Capacity with Polynomial Mapping). *Let $\phi_p(\cdot)$ be a polynomial mapping with degree at most $p$, and $\mathcal{M}$ be a matrix-valued memory that optimizes the internal objective of $\ell(\mathcal{M}_t; \phi_p(\boldsymbol{k}_t), \mathbf{v}_t) = \|\mathcal{M}_t \phi_p(\boldsymbol{k}_t) - \mathbf{v}_t\|_2^2$ with gradient descent. $\mathcal{M}$ can store the mapping of at most $\mathcal{O}(d_k^p)$ pairs of $(\boldsymbol{k}_i, \mathbf{v}_i)$ with linearly independent keys, where $d_k$ is the dimension of keys $\boldsymbol{k}_i$.*

Beyond the above intuition, polynomial kernels are further motivated by two perspectives: (1) Approximating Softmax using Taylor series; and (2) Input feature gating. For the sake of clarity, we continue with linear memory and two popular attentional biases i.e., $\ell^{(1)}(\mathcal{M}_t; \boldsymbol{k}_t, \mathbf{v}_t) = \langle \mathcal{M}_t \boldsymbol{k}_t, \mathbf{v}_t \rangle$ and $\ell^{(2)}(\mathcal{M}_t; \boldsymbol{k}_t, \mathbf{v}_t) = \|\mathcal{M}_t \phi(\boldsymbol{k}_t) - \mathbf{v}_t\|_2^2$. The same process can be applied on other attentional objectives and deep memory modules. Optimizing these objectives using gradient descent in the inner loop results in the following recurrent formulas:

$$\ell^{(1)}(\mathcal{M}_t; \boldsymbol{k}_t, \mathbf{v}_t) \; : \; \mathcal{M}_t = \mathcal{M}_{t-1} + \eta_t \mathbf{v}_t \phi(\boldsymbol{k}_t)^\top, \qquad \text{(Hebbian Rule)}$$

$$\ell^{(2)}(\mathcal{M}_t; \boldsymbol{k}_t, \mathbf{v}_t) \; : \; \mathcal{M}_t = \left(\mathbf{I} - \eta_t \phi(\boldsymbol{k}_t)\phi(\boldsymbol{k}_t)^\top\right)\mathcal{M}_{t-1} + \eta_t \mathbf{v}_t \phi(\boldsymbol{k}_t)^\top. \qquad \text{(Delta Rule)}$$

**Kernel Attention Perspective for the Special Case of Hebbian Rule.** The formulation for (Hebbian Rule) is equivalent to kernel linear attentions (Kacham et al., 2024b; Wang et al., 2025; Hua et al., 2022; Kasai et al., 2021; Katharopoulos et al., 2020; Arora et al., 2024). In this viewpoint, the role of $\phi(.)$ is to approximate `Softmax` or more accurately the exponential kernel. Since exponential kernel with normalization (i.e., `Softmax`) is not separable, it results in Transformers' quadratic time and memory complexity. However, Transformers' exponential feature map kernel ($\exp(\boldsymbol{q}_i^\top \boldsymbol{k}_j)$) can be approximated using its Taylor series as: $\exp\left(\boldsymbol{q}_i^\top \boldsymbol{k}_j\right) \approx 1 + \boldsymbol{q}_i^\top \boldsymbol{k}_j + \frac{(\boldsymbol{q}_i^\top \boldsymbol{k}_j)^2}{2!} + \frac{(\boldsymbol{q}_i^\top \boldsymbol{k}_j)^3}{3!} + \cdots$. Our polynomial feature map extends this approximation to a more general case of:

$$\exp\left(\boldsymbol{q}_i^\top \boldsymbol{k}_j\right) \approx \phi_p(\boldsymbol{q})\phi(\boldsymbol{k})^\top = a_0 + a_1 \boldsymbol{q}_i \boldsymbol{k}_j^\top + a_2(\boldsymbol{q}_i^\top \boldsymbol{k}_j)^2 + a_3(\boldsymbol{q}_i^\top \boldsymbol{k}_j)^3 + \cdots + a_p(\boldsymbol{q}_i^\top \boldsymbol{k}_j)^p, \quad (2)$$

with learnable parameters $a_i \in \mathbb{R}$ initialized at $a_i = \frac{1}{i!}$, the polynomial kernel can be viewed as an expressive approximator of `Softmax` attention. This provides theoretical motivation for using polynomial kernels, especially when memory capacity is limited; i.e., with (i) linear memory and (ii) Hebbian learning rule. This intuition, however, further generalizes to more expressive cases using deep memory modules and more complex attentional biases (i.e., Eq. Delta Rule). That is, $\exp(\cdot)$ feature mapping has infinite dimension and provides a more powerful similarity measure of keys and queries (i.e., $\boldsymbol{q}_i^\top \boldsymbol{k}_j$); however, its computation with normalization can cause additional memory and time complexity to the model. Using polynomial kernels in architectures with deep memory and complex attentional bias can further enhance performance by approximating more powerful representations for keys-queries similarities (i.e., $\boldsymbol{q}_i^\top \boldsymbol{k}_j$).

## 3.2 Long-term Memory with Context Memorization

As discussed earlier, one of the critical drawback of most existing recurrent models is their online nature, in which they optimize the inner objective (attentional bias) with respect to only the current input while retaining the previous state of the memory (Behrouz et al., 2025; Liu et al., 2024a), i.e.,

$$\min_{\mathcal{M}} \ell(\mathcal{M}; \boldsymbol{k}_t, \mathbf{v}_t) + \text{Ret}_t(\mathcal{M}, \mathcal{M}_{t-1}), \quad (3)$$

where $\text{Ret}(\cdot, \cdot)$ is the retention gate. This online nature while making the optimization of the memory simpler and faster, can cause sub-optimal memorization of the context as memory is greedily memorize individual tokens. In a more general case, however, one can optimize the memory at each time stamp with respect to the entire context (input sequence), i.e.,

$$\min_{\mathcal{M}} \sum_{i=1}^{t} \ell(\mathcal{M}; \boldsymbol{k}_i; \mathbf{v}_i). \quad (4)$$

This strict global formulation generally presents two critical limitations: (1) Efficiency: One of the important advantages of recurrent architectures is their efficiency at longer context in both training and inference. Optimizing the memory with respect to all the past tokens (entire context), however, (i) causes additional optimization constraints at each memory update step, resulting in inefficiency at extremely large sequences, and (ii) requires caching the past keys and values at the test time, increasing the memory consumption; (2) Context Pruning: In large context tasks optimizing with all past tokens can cause sub-optimal performance mainly due to the context change (or irrelevant context) in the middle of the input sequence. This observation has resulted to design architectures with retention (forget) gate, enabling models to erase memory when past context is no longer needed (Sun et al., 2023; Peng et al., 2025; Behrouz et al., 2024; 2025; Yang et al., 2024b).

To address these limitations, we present a sliding window recurrent model that optimizes its attentional bias w.r.t. a window of past tokens. For a memory module $\mathcal{M}(\cdot)$ and window length $c \geq 1$, we optimize the memory internal objective as:

$$\min_{\mathcal{M}} \sum_{i=t-c+1}^{t} \gamma_i^{(t)} \ell(\mathcal{M}; \boldsymbol{k}_i, \mathbf{v}_i), \quad (5)$$

where $\ell(\mathcal{M}; \boldsymbol{k}_i, \mathbf{v}_i)$ measures the predicted mapping for $(\boldsymbol{k}_i, \mathbf{v}_i)$ pair and $\gamma_i^{(t)}$ is the decay term for the effect of $i$-th token in the optimization process. Building upon this formulation, we present Omega rule, which is strictly more powerful than the popular Delta learning rule (Widrow & Hoff, 1988; Schlag et al., 2021):

**Omega Rule**: Let $\boldsymbol{k}_i \in \mathbb{R}^{d_k}$ and $\mathbf{v}_i \in \mathbb{R}^{d_v}$ be the input keys and values, and $\mathcal{M}(\cdot)$ be a neural architecture that serves as the memory module. Given a local context length of $c \in \mathbb{N}_{\geq 1}$, the updating the memory module $\mathcal{M}$ using Omega learning rule is defined as optimizing the following loss function with gradient descent:

$$\min_{\mathcal{M}} \sum_{i=t-c+1}^{t} \gamma_i^{(t)} \left\| \mathcal{M}(\boldsymbol{k}_i) - \mathbf{v}_i \right\|_2^2 \tag{6}$$

Following Behrouz et al. (2025), this update rule can be extended to $q$-Omega rule (or other variants) by replacing $\ell_2(\cdot)$ with $\ell_q(\cdot)$. In the extreme cases of (1) $c = 1$: the update rule becomes online (Delta rule); and (2) $c = \infty$ or context length: the update becomes global optimization w.r.t. all past tokens. In this formulation, parameters $\gamma_i^{(t)} \in [0, 1]$ act as hard (direct) gates for the past tokens. That is, $\gamma_i^{(t)} \to 0$ means that the model directly prunes the optimization of $i$-th token in the local context, while $\gamma_i^{(t)} \to 1$ means fully incorporating the optimization of memory for $i$-th token in the local context. In our design, we use input-dependent parameters for $\gamma_i^{(t)}$, providing in-context pruning ability. Note that, the design of sliding window recurrence allows such flexibility as for each token we need a constant number of gates; i.e., $\{\gamma_i^{(t)}\}_{i=1}^{c}$. Using input-dependent gates for the global optimization (Equation 4), however, can result in significant parameter increase and memory usage, diminishing the advantages of recurrent models.

**OMEGANET.** We now present OMEGANET, a novel sequence model that updates its memory using Omega rule. To enhance the memory capacity of OMEGANET, we use polynomial kernels on $\boldsymbol{k}$s and $\boldsymbol{q}$s. Accordingly, optimizing the objective in Equation 6, results in an update rule of OMEGANET as:

$$\mathcal{M}_t = \alpha_t \mathcal{M}_{t-1} - \nabla \underbrace{\sum_{i=t-c+1}^{t} \gamma_i^{(t)} \left\| \mathcal{M}(\phi(\boldsymbol{k}_i)) - \mathbf{v}_i \right\|_2^2}_{\text{Surprise of the context}}, \tag{7}$$

or in the spacial case of linear memory:

$$\mathcal{M}_t = \left( \text{diag}(\alpha_t) - \sum_{i=t-c+1}^{t} \gamma_i^{(t)} \phi(\boldsymbol{k}_i) \phi(\boldsymbol{k}_i)^\top \right) \mathcal{M}_{t-1} - \sum_{i=t-c+1}^{t} \gamma_i^{(t)} \mathbf{v}_i \phi(\boldsymbol{k}_i)^\top. \tag{8}$$

From the memory perspective, Omega rule (OMEGANET) does not measure the surprise of a token, but the surprise of a local context based on the context-aware combination of individual tokens within the context.

**Connection to Sliding Window Attention.** `Softmax` attention block can also be reformulated as a non-parametric solution to the $\ell_2(\cdot)$ regression with Nadaraya-Watson estimators (Zhang et al., 2022; Fan, 2018):

$$\mathcal{M}^* = \arg\min_{\mathcal{M}} \sum_{i=1}^{L} \mathbf{s}(\boldsymbol{k}_i, \boldsymbol{q}) \|\mathbf{v}_i - \mathcal{M}\|_2^2 = \sum_{i=1}^{L} \frac{\mathbf{s}(\boldsymbol{k}_i, \boldsymbol{q})}{\sum_{j=1}^{L} \mathbf{s}(\boldsymbol{k}_j, \boldsymbol{q})} \mathbf{v}_i, \tag{9}$$

where $L$ is the sequence length. While this formulation optimizes the memory $\mathcal{M}$ with respect to the entire sequence length, one can limit the optimization process to the past $c$ tokens, resulting in:

$$\mathcal{M}^* = \arg\min_{\mathcal{M}} \sum_{i=t-c+1}^{t} \mathbf{s}(\boldsymbol{k}_i, \boldsymbol{q}_i) \|\mathbf{v}_i - \mathcal{M}\|_2^2 = \sum_{i=t-c+1}^{t} \frac{\mathbf{s}(\boldsymbol{k}_i, \boldsymbol{q})}{\sum_{j=t-c+1}^{t} \mathbf{s}(\boldsymbol{k}_j, \boldsymbol{q})} \mathbf{v}_i, \tag{10}$$

which is equivalent to the sliding window attention (SWA). This connection provides an important insight on the difference of attention and recurrent models: Not only attention is a non-parametric solution (contrary to the parametric nature of recurrent models), it globally optimizes its internal objective (attentional bias), while all recent modern recurrent models are online learners (Sun et al., 2024; Yang et al., 2024a; Behrouz et al., 2024; 2025; Peng et al., 2025). Our formulations of sliding

window RNN and Omega rule close this gap by optimizing the memory with respect to a context window of past tokens based on parametric methods, effectively memorizing the context instead of individual tokens.

**Deep Linear Attention.** As a novel baseline, we present Deep (Gated) Linear Attention (DLA) that replaces a matrix-valued memory in (gated) linear attention (Katharopoulos et al., 2020; Yang et al., 2024b) with a deep neural network (e.g., $k$-layer MLP). As discussed earlier in (Hebbian Rule), using dot product similarity as the internal attentional bias results in linear attention. Thus, leveraging recent deep memory modules (Sun et al., 2024; Behrouz et al., 2024; 2025), we optimize the memory using gradient descent with dot product attentional bias:

$$\mathcal{M}_t = \alpha_t \mathcal{M}_{t-1} - \eta_t \nabla \ell(\mathcal{M}_{t-1}; \phi(\boldsymbol{k}_t), \mathbf{v}_t), \tag{11}$$

where $\ell(\mathcal{M}_{t-1}; \phi(\boldsymbol{k}_t), \mathbf{v}_t) = \langle \mathcal{M}_{t-1}(\phi(\boldsymbol{k}_t)), \mathbf{v}_t \rangle$ and $\phi(\cdot)$ is a polynomial kernel. The training of DLA can simply be parallelized using the hybrid of linear and non-linear chunk-wise training, the same as Sun et al. (2024); Behrouz et al. (2024) and our discussion in Appendix H.

**Sliding Window Linear Attention.** Building upon the above intuition and the connection of our formulation to SWA, we present Sliding Window Linear Attention (SWLA) block. Following the formulation of linear attention in associative memory perspective (Behrouz et al., 2025), we use dot product similarity (i.e., $\ell(\mathcal{M}_t; \boldsymbol{k}_i, \mathbf{v}_i) = \langle \mathcal{M}_t(\boldsymbol{k}_i), \mathbf{v}_i \rangle$) as the attentional bias and optimize the loss function using gradient descent. For the sake of clarity, we use a linear memory here to derive the closed form:

$$\mathcal{M}_t = \alpha_t \mathcal{M}_{t-1} - \eta_t \nabla \sum_{i=t-c+1}^{t} \ell(\mathcal{M}_{t-1}; \phi(\boldsymbol{k}_i), \mathbf{v}_i) = \mathcal{M}_{t-1} + \sum_{i=t-c+1}^{t} \gamma_i^{(t)} \mathbf{v}_i \phi(\boldsymbol{k}_i)^\top \tag{12}$$

In the online case ($c = 1$) and $\phi(\cdot) = (\cdot)$, this recurrence is the same as linear attention (Katharopoulos et al., 2020).

# 4 ATLAS: A LOCALLY OPTIMAL MEMORY WITH HIGH CAPACITY

Although the design of Omega rule allows the model to memorize the context instead of individual tokens and also the use of polynomial (or exponential) feature mapping increases memory capacity, the memory management (i.e., optimization of mappings between keys and values) is still limited to a simple gradient descent. This choice of optimizer can lead the model to a low-quality solution at a local optima, damaging the performance of the model in longer contexts. To overcome this issue, we suggest using Muon optimizer (Jordan et al., 2024) (with weight decay) that not only approximates second-order information, but it also mostly leverages matrix multiplication and can be parallelized across the sequence. Accordingly, the use of Muon for optimizing the internal objective in Equation 6, results in the following update rule:

$$\mathcal{M}_t = \alpha_t \mathcal{M}_{t-1} - \eta_t \, \texttt{NewtonShulz-}k(\mathcal{S}_t), \tag{13}$$

$$\mathcal{S}_t = \theta_t \mathcal{S}_{t-1} + \nabla \sum_{i=t-c+1}^{t} \gamma_i^{(t)} \left\| \mathcal{M}\left(\phi^*(\boldsymbol{k}_i)\right) - \mathbf{v}_i \right\|_2^2, \tag{14}$$

where $c$ is the local context length and $k$ is the number steps for `NewtonShulz` operations. For the additional discussion on the algorithm and this operation we refer the reader to Jordan et al. (2024). Following the literature on Muon optimizer, we know that when $k \to \infty$, then `NewtonShulz-`$k(\mathcal{S}_t)$ converges to the nearest semi-orthogonal matrix to the momentum term $\mathcal{S}_t$ and so approximate second-order information with a lower error. Therefore, interestingly, parameter $k$ can be considered as an internal test-time compute parameter in ATLAS, where using more steps can potentially result in better memorization. In Appendix H we show that ATLAS and Omega rule are parallelizable without any significant computational overhead.

**Architectural Backbone.** As for the architectural backbone, we follow the recent modern recurrent models (Behrouz et al., 2024; Arora et al., 2024; Yang et al., 2024c; Allen-Zhu, 2025) and use linear layers to project keys, values, and queries, followed by short convolution layers with size 4. We apply normalization on keys and queries to stabilize the training. We also follow Behrouz et al.

Table 1: Performance of ATLAS and baselines on language modeling and common-sense reasoning tasks. Hybrid models are marked with *. The best results are highlighted highlighted .

| Model | Wiki. ppl ↓ | LMB. ppl ↓ | LMB. acc ↑ | PIQA acc ↑ | Hella. acc_n ↑ | Wino. acc ↑ | ARC-e acc ↑ | ARC-c acc_n ↑ | SIQA acc ↑ | BoolQ acc ↑ | Avg. ↑ |
|---|---|---|---|---|---|---|---|---|---|---|---|
| | | | | | 760M params / 30B tokens | | | | | | |
| Transformer++ | 25.21 | 27.64 | 35.8 | 66.9 | 42.2 | 51.9 | 60.4 | 32.5 | 39.5 | 60.4 | 48.69 |
| RetNet | 26.08 | 24.45 | 34.5 | 67.2 | 41.6 | 52.1 | 63.2 | 32.8 | 38.4 | 57.9 | 48.46 |
| DeltaNet | 24.37 | 24.60 | 37.1 | 66.9 | 42.0 | 50.7 | 64.9 | 31.4 | 39.9 | 59.0 | 48.97 |
| Gated DeltaNet | 21.18 | 22.09 | 35.5 | 68.0 | 44.9 | 50.7 | 66.9 | 33.1 | 39.2 | 59.1 | 49.69 |
| Samba* | 20.63 | 22.71 | 39.7 | 69.2 | 47.4 | 52.0 | 66.9 | 33.2 | 39.0 | 61.2 | 51.08 |
| Gated DeltaNet-H2* | 19.88 | 20.83 | 39.2 | 69.0 | 48.2 | 52.6 | 67.0 | 35.5 | 39.4 | 61.1 | 51.49 |
| Titans (LMM) | 20.04 | 21.96 | 37.4 | 69.3 | 48.5 | 52.3 | 66.3 | 35.8 | 40.1 | 62.8 | 51.56 |
| MEMORA | 22.28 | 22.31 | 38.2 | 67.8 | 49.3 | 53.3 | 63.6 | 36.1 | 40.9 | 63.0 | 51.52 |
| SWDT (ours) | 19.89 | 21.52 | 36.2 | 68.3 | 45.2 | 53.0 | 65.4 | 34.2 | 39.5 | 59.5 | 50.1 |
| DLA (ours) | 23.12 | 22.09 | 36.1 | 68.0 | 47.9 | 52.7 | 65.8 | 34.6 | 39.1 | 59.6 | 50.46 |
| OMEGANET (ours) | 19.16 | 20.14 | 38.7 | 69.8 | 50.0 | 53.3 | 67.8 | 36.8 | 39.6 | 64.4 | 52.56 |
| ATLAS (ours) | 18.92 | 21.01 | 39.1 | 69.7 | 50.2 | 53.5 | 67.5 | 37.1 | 40.7 | 64.3 | 52.77 |
| ATLAS++ (ours) | 19.04 | 20.03 | 39.7 | 69.7 | 51.1 | 53.2 | 68.2 | 37.4 | 40.9 | 64.4 | 53.09 |
| ATLAS (MAG) | 18.62 | 21.18 | 40.0 | 70.3 | 50.5 | 53.0 | 68.1 | 36.5 | 41.2 | 65.0 | 53.08 |
| ATLAS (MAL) | 19.07 | 21.46 | 38.8 | 69.2 | 50.5 | 53.6 | 67.3 | 36.1 | 41.0 | 64.5 | 52.63 |
| | | | | | 1.3B params / 100B tokens | | | | | | |
| Transformer++ | 18.53 | 18.32 | 42.6 | 70.0 | 50.2 | 53.5 | 68.8 | 35.1 | 40.7 | 57.1 | 52.25 |
| RetNet | 19.08 | 17.27 | 40.5 | 70.1 | 49.2 | 54.1 | 67.3 | 33.8 | 40.8 | 60.4 | 52.02 |
| DeltaNet | 17.71 | 16.88 | 42.5 | 70.7 | 50.9 | 53.3 | 68.5 | 35.7 | 40.2 | 55.3 | 52.14 |
| Gated DeltaNet | 16.42 | 12.17 | 46.6 | 72.2 | 55.8 | 57.4 | 71.2 | 38.4 | 40.6 | 60.2 | 55.32 |
| Samba* | 16.13 | 13.29 | 44.9 | 70.9 | 53.4 | 55.6 | 68.8 | 36.2 | 40.0 | 62.1 | 54.00 |
| Gated DeltaNet-H2* | 15.91 | 12.55 | 48.8 | 72.2 | 56.9 | 57.8 | 71.4 | 39.1 | 41.2 | 61.6 | 56.18 |
| Titans (LMM) | 15.60 | 11.41 | 49.1 | 73.1 | 56.3 | 59.8 | 72.4 | 40.8 | 42.1 | 61.0 | 56.82 |
| MEMORA | 15.90 | 12.04 | 48.7 | 73.1 | 56.0 | 57.4 | 71.5 | 37.9 | 40.2 | 61.3 | 55.87 |
| OMEGANET (ours) | 14.91 | 11.26 | 49.7 | 73.4 | 57.6 | 59.7 | 72.6 | 40.3 | 42.4 | 62.1 | 57.23 |
| ATLAS (ours) | 14.97 | 10.98 | 50.1 | 73.9 | 57.3 | 60.2 | 72.8 | 41.0 | 42.9 | 62.8 | 57.62 |
| ATLAS++ (ours) | 14.40 | 10.72 | 50.8 | 73.5 | 59.4 | 61.1 | 71.3 | 43.7 | 42.5 | 61.9 | 58.03 |

(2024) and use two hybrid variants of MAL and MAG for our ATLAS model. The architectures are illustrated in Figure 3. For models with deep memory architectures we use 2-layer MLP with residual connections $\mathcal{M}(\cdot) = (\cdot) + W_1 \sigma(W_2(\cdot))$. We further extend this memory architecture, which is commonly used in recent studies (Behrouz et al., 2024; Irie et al., 2021; Behrouz et al., 2025), to gated MLP layer as $\mathcal{M}(\cdot) = (\cdot) + W_1 \left( \sigma \left( W_2(\cdot) \right) \otimes W_3(\cdot) \right)$, where $W_1, W_2, W_3$ are linear learnable matrices. We refer to ATLAS with the above memory architecture as ATLAS++.

## 5 EXPERIMENTS

Next, we evaluate the performance of ATLAS, and OMEGANET in language modeling, common-sense reasoning, needle in haystack, and in-context recall tasks. Additional experimental results are in Appendix J.

**Setup.** We train our models with training context window of size 4K using FineWeb dataset (Penedo et al., 2024). We use model size of 340M, 400M, 790M, and 1.3B parameters and train them on 15B, 15B, 30B, and 100B tokens sampled from the dataset. Baseline results are reported by Yang et al. (2024a); Behrouz et al. (2024; 2025). Perplexity is measured on held-out validation data. As for the downstream tasks, we evaluate trained models on Wikitext (Merity et al., 2017), LMB (Paperno et al., 2016), PIQA (Bisk et al., 2020), HellaSwag (Zellers et al., 2019), WinoGrande (Sakaguchi et al., 2021), ARC-easy (ARC-e) and ARC-challenge (ARC-c) (Clark et al., 2018), SIQA (Sap et al., 2019), and BoolQ (Clark et al., 2019). Additional details about the experimental setups and other used datasets are in Appendix I.

### 5.1 LANGUAGE MODELING AND COMMON-SENSE REASONING

The results for ATLAS, and OMEGANET as well as their corresponding baselines of SWDT and DLA with the size of 760M and 1.3B are reported in Table 1. (see Appendix J for the full results). Among non-hybrid models, including Transformer++, our ATLAS, and OMEGANET achieve the best performance in both perplexity and accuracy measures. We attribute this performance to their

ability to memorize the context rather than individual tokens. Comparing OMEGANET with Titans, that also uses the same momentary objective (i.e., $\ell_2$ loss), but with context window of 1, we can observe the effectiveness of having non-online learning rule. On the other hand, our models, alone without any attention, can outperform hybrid models, while their hybrid variant of MAG further improve their performance. This performance gain is also related to the use of polynomial kernels that enhance the memory capacity of the model. See Figure 2 for a more controlled study on the effect of different components.

## 5.2 LONG CONTEXT: NEEDLE IN A HAYSTACK

One of our main motivations to design ATLAS is to enhance the performance of long-term neural memory module in long context tasks. Accordingly, to evaluate the effectiveness of our designs for improving the effective context length and memory capacity, we perform an experiment on needle-in-haystack tasks of RULER (Hsieh et al., 2024) benchmark. The performance of ATLAS and its hybrid variants, as well as our Transformer-like architectures and baselines are reported in Table 2.

Table 2: Performance of ATLAS and baselines on S-NIAH task from RULER benchmark. The best results among simple and hybrid models are highlighted.

| Model | S-NIAH-PK | | | | S-NIAH-N | | | | S-NIAH-W | | |
|---|---|---|---|---|---|---|---|---|---|---|---|
| | 2K | 4K | 8K | 16K | 2K | 4K | 8K | 16K | 2K | 4K | 8K |
| TTT | 98.4 | 98.8 | 98.0 | 88.4 | 60.2 | 36.6 | 10.2 | 4.4 | 78.8 | 28.0 | 4.4 |
| DeltaNet | 96.8 | 98.8 | 98.6 | 71.4 | 47.2 | 15.4 | 12.8 | 5.4 | 46.2 | 20.0 | 1.6 |
| Titans (LMM) | 99.8 | 98.4 | 98.2 | 96.2 | 100.0 | 99.8 | 93.4 | 80.2 | 90.4 | 89.4 | 85.8 |
| ATLAS | 100 | 99.2 | 98.0 | 97.0 | 100.0 | 100.0 | 93.0 | 84.0 | 93.2 | 90.6 | 86.2 |
| Samba | 98.8 | 98.0 | 97.4 | 97.2 | 98.8 | 98.6 | 96.2 | 95.6 | 96.8 | 90.0 | 84.0 |
| Gated DeltaNet-H2* | 99.2 | 97.8 | 97.4 | 98.4 | 98.0 | 97.8 | 96.2 | 95.8 | 97.4 | 96.8 | 88.4 |
| ATLAS (MAG) | 100 | 100 | 99.4 | 98.6 | 100 | 99.2 | 97.4 | 97.0 | 99.4 | 98.2 | 92.4 |
| ATLAS (MAL) | 99.8 | 99.6 | 98.4 | 96.8 | 99.8 | 98.0 | 97.2 | 96.8 | 98.0 | 98.4 | 92.6 |

ATLAS shows very good performance compared to the recurrent baselines, outperforming modern recurrent neural networks such as Titans and DeltaNet. Its hybrid variants further improve its effective context length, effectively extrapolating to sequences with $\times 4$ of their training context size. We attribute this performance to the proposed enhancements for the capacity of the memory. We further perform ablation studies to validate this claim. Also, our Transformer-like architectures outperforms the baselines, even our hybrid variants of ATLAS in longer contexts. This shows the importance of exponential feature mapping in longer sequences.

## 5.3 ABLATION STUDY

In this section, we perform an ablation study on the different components of ATLAS. The results for ablation study are reported in Figure 2. The results show that: (1) more powerful memory architectures such as gated MLP can further enhance the performance of ATLAS; (2) The hybrid variants further improve the performance, where MAG shows better improvement compared to MAL architecture; (3) Polynomial mappings as well as deep memory are particularly important when we use context memorization (i.e., Omega rule).

Figure 2: Ablation Study on ATLAS. All components of ATLAS are positively contributing to its performance.

| Model | Language Modeling ppl ↓ | C.S. Reasoning acc ↑ |
|---|---|---|
| ATLAS | 19.97 | 52.77 |
| +Gated MLP Memory | 19.53 | 53.09 |
| +Attn (MAG) | 19.90 | 53.08 |
| +Attn (MAL) | 20.26 | 52.63 |
| Linear Memory | 21.03 | 49.74 |
| w/o Muon | 19.65 | 52.56 |
| $c = 1$ | 21.98 | 49.26 |
| w/o Polynomial Mapping | 22.14 | 50.57 |

## 6 CONCLUSION

We introduced ATLAS, a new long-term memory module designed to address the core limitations of modern recurrent models in long-context understanding: limited memory capacity, online-only updates, and weak memory management. Our proposed sliding window learning rule, higher-order feature mappings, and advanced memory optimizers offer a principled and scalable approach to overcoming these challenges. Empirically, our models—OMEGANET, and ATLAS—achieve consistent improvements over modern RNN variants across diverse benchmarks, and also closer the gap with Transformers in in-context retrieval tasks. Theoretically, we provided insight into memory capacity and optimization dynamics, offering explanations for the context length limitations observed in prior works.

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

Table 3: A summary of the recent modern recurrent neural networks. We compare these architectures based on five characteristics: (1) Dynamic decay; (2) Deep neural memory; (3) non-linear memory capacity; (4) Locally optimal: managing memory by (approximating) the second-order information about tokens; (5) Flexible context: the ability to flexibly memorize the context. $\phi(\cdot)$ and $\phi^*(\cdot)$ represent polynomial and infinite-dimensional feature mappings (see Equation 36).

| Model | Attentional Bias $\ell(\cdot;\cdot)$ | Optimizer | Dynamic Decay | Deep Memory | Non-linear Capacity† | Locally Optimal | Flexible Context | Memory Write Operation |
|---|---|---|---|---|---|---|---|---|
| Attention | $\sum_{t=1}^{L} a_t\|\mathcal{M}\boldsymbol{k}_t - \mathbf{v}_t\|_2^2$ | NP‡ | ✗ | ✗ | ✓ | ✓ | ✗ | $\mathcal{M}_t = \mathcal{M}_{t-1} \cup \{(\boldsymbol{k}_t, \mathbf{v}_t)\}$ |
| SWA | $\sum_{t=c}^{L} a_t\|\mathcal{M}\boldsymbol{k}_t - \mathbf{v}_t\|_2^2$ | NP | ✗ | ✗ | ✓ | ✓ | ✓ | $\mathcal{M}_t = (\mathcal{M}_{t-1} \setminus \{(\boldsymbol{k}_c, \mathbf{v}_c)\}) \cup \{(\boldsymbol{k}_t, \mathbf{v}_t)\}$ |
| Linear Attention | $\langle \mathcal{M}_t \boldsymbol{k}_t, \mathbf{v}_t \rangle$ | GD | ✗ | ✗ | ✗ | ✗ | ✗ | $\mathcal{M}_t = \mathcal{M}_{t-1} + \mathbf{v}_t \boldsymbol{k}_t^\top$ |
| RetNet | $\langle \mathcal{M}_t \boldsymbol{k}_t, \mathbf{v}_t \rangle$ | GD | ✗ | ✗ | ✗ | ✗ | ✗ | $\mathcal{M}_t = \alpha_t \mathcal{M}_{t-1} + \mathbf{v}_t \boldsymbol{k}_t^\top$ |
| GLA | $\langle \mathcal{M}_t \boldsymbol{k}_t, \mathbf{v}_t \rangle$ | GD | ✓ | ✗ | ✗ | ✗ | ✗ | $\mathcal{M}_t = \text{Diag}(\alpha_t)\mathcal{M}_{t-1} + \mathbf{v}_t \boldsymbol{k}_t^\top$ |
| PolySketchFor. | $\langle \mathcal{M}_t \boldsymbol{k}_t^p, \mathbf{v}_t \rangle$ | GD | ✗ | ✗ | ✓ | ✗ | ✗ | $\mathcal{M}_t = \mathcal{M}_{t-1} + \mathbf{v}_t (\boldsymbol{k}_t^\top)^p$ |
| TTT | $\|\mathcal{M}_t(\boldsymbol{k}_t) - \mathbf{v}_t\|_2^2$ | GD | ✗ | ✓ | ✗ | ✗ | ✗ | $\mathcal{M}_t = \mathcal{M}_{t-1} - \eta \nabla\ell(\mathcal{M}_{t-1}; \boldsymbol{k}_t, \mathbf{v}_t)$ |
| DeltaNet | $\|\mathcal{M}_t \boldsymbol{k}_t - \mathbf{v}_t\|_2^2$ | GD | ✗ | ✗ | ✗ | ✗ | ✗ | $\mathcal{M}_t = (\mathbf{I} - \beta_t \boldsymbol{k}_t \boldsymbol{k}_t^\top)\mathcal{M}_{t-1} + \beta_t \mathbf{v}_t \boldsymbol{k}_t^\top$ |
| Longhorn | $\|\mathcal{M}_t \boldsymbol{k}_t - \mathbf{v}_t\|_2^2$ | Implicit GD | ✗ | ✗ | ✗ | ✗ | ✗ | $\mathcal{M}_t = (\mathbf{I} - \delta_t \boldsymbol{k}_t \boldsymbol{k}_t^\top)\mathcal{M}_{t-1} + (\delta_t \odot \mathbf{v}_t)\boldsymbol{k}_t$ § |
| Gated DeltaNet | $\|\mathcal{M}_t \boldsymbol{k}_t - \mathbf{v}_t\|_2^2$ | GD | ✓ | ✗ | ✗ | ✗ | ✗ | $\mathcal{M}_t = \alpha_t(\mathbf{I} - \beta_t \boldsymbol{k}_t \boldsymbol{k}_t^\top)\mathcal{M}_{t-1} + \beta_t \mathbf{v}_t \boldsymbol{k}_t^\top$ |
| RWKV-7 | $\|\mathcal{M}_t \boldsymbol{k}_t - \mathbf{v}_t\|_2^2$ | GD | ✓ | ✗ | ✗ | ✗ | ✗ | $\mathcal{M}_t = (\text{diag}(\alpha_t) - \beta_t \boldsymbol{k}_t \boldsymbol{k}_t^\top)\mathcal{M}_{t-1} + \beta_t \mathbf{v}_t \boldsymbol{k}_t^\top$ |
| Titans | $\|\mathcal{M}_t(\boldsymbol{k}_t) - \mathbf{v}_t\|_2^2$ | GD w/ M.* | ✓ | ✓ | ✗ | ✗ | ✗ | $\mathcal{M}_t = \alpha_t \mathcal{M}_{t-1} + \mathcal{S}_t$ $\mathcal{S}_t = \eta_t \mathcal{S}_{t-1} - \theta_t \nabla\ell(\mathcal{M}_{t-1}; \boldsymbol{k}_t, \mathbf{v}_t)$ |
| Titans– | $\|\mathcal{M}_t(\boldsymbol{k}_t) - \mathbf{v}_t\|_2^2$ | GD | ✓ | ✓ | ✗ | ✗ | ✗ | $\mathcal{M}_t = \alpha_t \mathcal{M}_{t-1} - \eta_t \nabla\ell(\mathcal{M}_{t-1}; \boldsymbol{k}_t, \mathbf{v}_t)$ |
| MONETA | $\|\mathcal{M}_t(\boldsymbol{k}_t) - \mathbf{v}_t\|_p^p$ | GD | ✓ | ✓ | ✗ | ✗ | ✗ | $\mathcal{M}_t = \alpha_t \mathcal{M}_{t-1} - \eta_t \nabla\ell(\mathcal{M}_{t-1}; \boldsymbol{k}_t, \mathbf{v}_t)$ |
| **Our Models** | | | | | | | | |
| DLA | $\langle \mathcal{M}_t(\phi(\boldsymbol{k}_t)), \mathbf{v}_t \rangle$ | GD | ✓ | ✓ | ✗ | ✗ | ✗ | $\mathcal{M}_t = \alpha_t \mathcal{M}_{t-1} - \eta_t \nabla\ell(\mathcal{M}_{t-1}; \boldsymbol{k}_t, \mathbf{v}_t)$ |
| SWDT | $\sum_{i=c}^{L} \langle \mathcal{M}_t(\phi^*(\boldsymbol{k}_i)), \mathbf{v}_i \rangle$ | GD | ✓ | ✓ | ✓ | ✗ | ✓ | $\mathcal{M}_t = \alpha_t \mathcal{M}_{t-1} - \eta_t \nabla\ell(\mathcal{M}_{t-1}; \boldsymbol{k}_t, \mathbf{v}_t)$ |
| OmegaNet | $\sum_{i=c}^{L} \gamma_i \|\mathcal{M}_t(\phi(\boldsymbol{k}_i)) - \mathbf{v}_i\|_2^2$ | GD | ✓ | ✓ | ✓ | ✗ | ✓ | $\mathcal{M}_t = \alpha_t \mathcal{M}_{t-1} - \eta_t \nabla\ell(\mathcal{M}_{t-1}; \boldsymbol{k}_t, \mathbf{v}_t)$ |
| ATLAS | $\sum_{i=c}^{L} \gamma_i \|\mathcal{M}_t(\phi(\boldsymbol{k}_i)) - \mathbf{v}_i\|_2^2$ | Muon | ✓ | ✓ | ✓ | ✓ | ✓ | $\mathcal{M}_t = \alpha_t \mathcal{M}_{t-1} - \eta_t \text{ NS-5}(\mathcal{S}_t)$ $\mathcal{S}_t = \theta_t \mathcal{S}_{t-1} - \nabla\ell(\mathcal{M}_{t-1}; \boldsymbol{k}_t, \mathbf{v}_t)$ |

† The matrix-valued memory version is considered.  ‡ NP: Nonparametric  § $\delta_t = \frac{\beta_t}{1+\beta_t \boldsymbol{k}_t^\top \boldsymbol{k}_t}$.  * Gradient Descent with Momentum.

## A MOTIVATION: MEMORY PERSPECTIVE

Associative memory—the ability to map different entities or events—is an inseparable component of learning in humans (Terry, 2017) and so has motivated several recent studies to understand the state-of-the-art deep learning architectures through its lens (Ramsauer et al., 2021; Behrouz et al., 2024; 2025; Wang et al., 2025). In this perspective, memory is defined as a neural update caused by an input; the more surprising the input is, the more it affects the memory and so is memorable. Therefore, finding an effective "surprise metric" is a critical step towards designing such memory modules. As earlier discussed by Behrouz et al. (2025; 2024), almost all existing architectures use a surprise metric that updates the memory based on the current input. An event (as a sequence of tokens), however, might not consistently be surprising through a long-period of time although it is memorable. To overcome this issue, Behrouz et al. (2024) suggest breaking the surprise metric into two parts of "momentary" and "past" surprise, incorporating the *cumulative* surprise of past inputs when updating the memory with respect to the current input. This design, however, can miss the *context* by memorizing individual tokens. To this end, in this work, we present a long-term neural memory module that measures the surprise of a local (or global) context window, meaning that it learns how to memorize the (~~token~~) context at test time.

Through the paper, we use the terminology "Test Time Memorization" because the process involves storing and retrieving information strictly within the global context, without updating the model's core learned parameters (i.e., outer-loop) or initial states from pre-training. Typically, no persistent learning or skill acquisition carries over to new, independent global context once the memory is cleared. Thus, we prefer the use of "test time memorization" over using "test time training".

## B ADDITIONAL RELATED WORK

**Modern Linear Recurrent Neural Networks**[1]. Recent research endeavors have concentrated on mitigating the quadratic computational complexity and inherent limitations of Transformer mod-

---

[1]Note that here the term "linear" refers to their fast training and inference procedures. This does not refer to their recurrence formula as some models like Titans (Behrouz et al., 2024), YAAD, MONETA, MEMORA (Behrouz et al., 2025), and TTT (Sun et al., 2024) are based on *non-linear* recurrence but fast at training and inference.

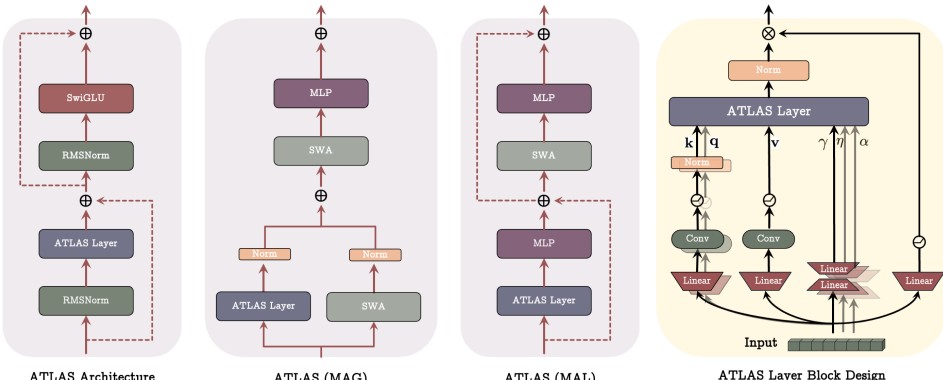

Figure 3: Visualization of the ATLAS's (and our other variants') architecture, and its hybrid counterpart with SWA.

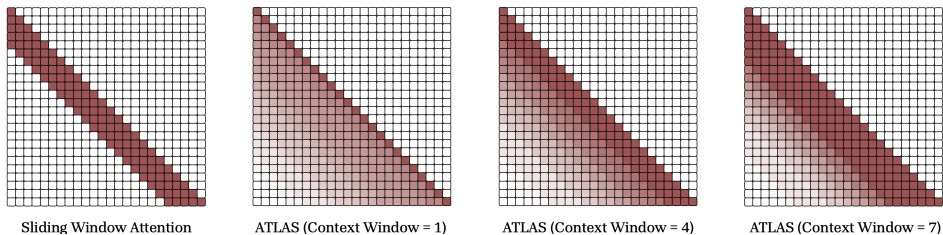

Figure 4: The illustration of tokens dependencies in SWA and ATLAS or OMEGANET with different context length.

els in processing long-context sequences. This has led to the development of efficient recurrent alternatives, primarily motivated by their rapid inference and training capabilities (Tiezzi et al., 2024). Initial advancements in this domain, exemplified by models such as RetNet (Sun et al., 2023), RWKV (Peng et al., 2023), and S5 (Smith et al., 2023), employed data-independent transition matrices coupled with Hebbian-like update mechanisms. Subsequently, a second generation of models emerged, incorporating input-dependent parameters within these linear architectures (e.g., linear RNNs (Hasani et al., 2023; Smith et al., 2023), RWKV6 (Peng et al., 2024)). These models also explored more expressive memory updating rules, notably those based on the delta rule (Peng et al., 2025; Schlag et al., 2021; Yang et al., 2024c;a; Liu et al., 2024a). Further evolution in this line of research has extended these memory architectures to deeper models, while concurrently utilizing delta-rule-like update mechanisms (Sun et al., 2024) or data-dependent momentum-based update rules with forget gating (Behrouz et al., 2024). More recently, to augment the performance of delta-rule-based sequential models, Siems et al. (2025) have proposed the application of multiple gradient descent updates per token, thereby yielding more expressive sequence models, particularly in state tracking tasks. In addition to the above fast linear recurrent sequence models, several studies have focused on RNNs with non-linear recurrence (Behrouz et al., 2025; Csordás et al., 2024; Merrill et al., 2024; Lim et al., 2024; Schöne et al., 2025; Karami & Mirrokni, 2025; Von Oswald et al., 2023; Gonzalez et al., 2024), and how their training can be faster (Gonzalez et al., 2024; Lim et al., 2024; Schöne et al., 2025).

**Fast Weight Programs.** The conceptualization of linear layers as key-value associative memory systems can be traced back to Hopfield networks (Hopfield, 1982). This concept was subsequently developed in the context of fast weight programmers, wherein dynamic fast programs are integrated into recurrent neural networks to serve as writable memory stores (Schlag et al., 2021; Schmidhuber, 1992; 1993). Among the learning paradigms for such systems, Hebbian learning (Hebb, 2005) and the delta rule (Prados & Kak, 1989) have emerged as the most prominent. Both learning rules have been the subject of extensive investigation within the existing literature (Munkhdalai & Yu, 2017; Schmidhuber, 1992; Munkhdalai et al., 2019; Schlag et al., 2021; Irie et al., 2021; Yang et al., 2024c;a).

**Hopfield Networks.** Our formulation is architecturally founded upon the broad concept of associative memory, wherein the primary objective is to learn an underlying mapping between keys and values. Seminal work by Hopfield (1982) on Hopfield Networks introduced one of the earliest neural architectures explicitly based on associative memory, defining it through the minimization of an energy function for storing key-value pairs. Although traditional Hopfield networks have seen diminished applicability in recent years, primarily due to constraints in vector-valued memory capacity and the nature of their energy function, several contemporary studies have focused on enhancing their capacity through various methodologies. These include efforts by Krotov (2021), Li et al. (2024b), and Krotov & Hopfield (2016). Notably, extensions to the energy function of these models, often incorporating exponential kernels, have been explored (Krotov & Hopfield, 2016; Lucibello & Mézard, 2024). Furthermore, the relationship between these modernized Hopfield networks and Transformer architectures has been a subject of recent investigation (Ramsauer et al., 2021; Hu et al., 2024).

## C  BACKGROUNDS

**Attention.** Attention is a critical component of Transformers that acts as their associative memory (Bietti et al., 2023; Sun et al., 2024; Behrouz et al., 2025). Given input $x \in \mathbb{R}^{N \times d_{\text{in}}}$, causal attention computes output $\mathbf{y} \in \mathbb{R}^{N \times d_{\text{in}}}$ over input dependent key, value, and query matrices $\mathbf{Q} = x\mathbf{W_Q}, \mathbf{K} = x\mathbf{W_K}$, and $\mathbf{V} = x\mathbf{W_V}$ as:

$$\mathbf{y}_i = \sum_{j=1}^{i} \frac{\exp\left(\mathbf{q}_i^\top \mathbf{k}_j / \sqrt{d_{\text{in}}}\right) \mathbf{v}_j}{\sum_{\ell=1}^{i} \exp\left(\mathbf{q}_i^\top \mathbf{k}_\ell / \sqrt{d_{\text{in}}}\right)} = \frac{1}{Z_i} \sum_{j=1}^{i} \exp\left(\mathbf{q}_i^\top \mathbf{k}_j / \sqrt{d_{\text{in}}}\right) \mathbf{v}_j, \tag{15}$$

where $\mathbf{W_Q}, \mathbf{W_K}$, and $\mathbf{W_V} \in \mathbb{R}^{d_{\text{in}} \times d_{\text{in}}}$ are learnable parameters, and $Z_i = \sum_{\ell=1}^{i} \exp\left(\mathbf{q}_i^\top \mathbf{k}_\ell / \sqrt{d_{\text{in}}}\right)$ is the normalization term. Despite Transformers' simple parallelizable training and effectiveness in recall-intensive tasks (Arora et al., 2024), their generation process and long-context scaling are significant drawbacks, as attention requires at least $N \times d$ operations per token to calculate the output (see Equation 15). Therefore, in recent years, there have been an extensive research effort to design alternative architectures. We divide and review these studies into two groups: (1) Linear shallow memory recurrent models, (2) Deep memory modules:

**(Linear) Recurrent Models.** Linear RNNs have recently gained attention as efficient Transformer alternatives due to their parallelizable, linear-time training and comparable performance (Sun et al., 2023; Peng et al., 2023). Early modern RNN variants, often based on Hebbian (Hebb, 2005) or Delta (Widrow & Hoff, 1988) learning rules, compress data into vector-valued or matrix-valued memory (Katharopoulos et al., 2020; Sun et al., 2023; Kacham et al., 2024b; Liu et al., 2024a; Schlag et al., 2021; Lim et al., 2024). Let $\mathcal{M}_t \in \mathbb{R}^{d \times n}$ be the memory (where $n = 1$ yields vector-valued memory), and $\boldsymbol{k}, \mathbf{v} \in \mathbb{R}^d$ be the keys and values (projections of input $x_t \in \mathbb{R}^d$). A simple general formulation for such linear RNNs is:

$$\mathcal{M}_t = A_t * \mathcal{M}_{t-1} + \mathbf{v}_t \boldsymbol{k}_t^\top, \tag{16}$$

where $*$ is an arbitrary associative operator and $A_t$ is a data-(in)dependent diagonal or low-rank plus identity matrix (Yang et al., 2024c). Despite the efficient *linear* recurrent nature of these models, their memory can overflow, particularly with increasing context length. Although forget gates have recently significantly improved memory management in these architectures (Sun et al., 2023; Peng et al., 2025), their memory's expressivity remains bounded by its linear structure.

## D  MIRAS FRAMEWORK

As discussed earlier, Behrouz et al. (2025) formalized the concept of associative memory as:

**Definition 2** (Behrouz et al. (2025))**.** *Given a set of keys $\mathcal{K} \subseteq \mathbb{R}^{d_k}$ and values $\mathcal{V} \subseteq \mathbb{R}^{d_v}$, associative memory is an mapping $\mathcal{M} : \mathcal{K} \to \mathcal{V}$. Learning the associative memory is based on an objective $\mathcal{L}$, called* Attentional Bias*, that determines the type of memory and its priorities:*

$$\mathcal{M}^* = \arg\min_{\mathcal{M}} \quad \mathcal{L}(\mathcal{M}(\mathcal{K}); \mathcal{V}). \tag{17}$$

Optimizing this objective using an iterative algorithm (e.g., gradient descent) results in the memory update rule. Thus, the sequence model is a meta in-context learner with two optimization levels:

1. Inner Loop: Where parameters of the memory module are optimized (i.e., $\boldsymbol{\theta}_{\mathcal{M}} = \{W_1, W_2, \ldots, W_{\mathcal{L}_{\mathcal{M}},\ldots}\}$). In the inner optimization loop, all other parameters from the model are considered hyperparameters and are fixed and *not* optimized.

2. Outer Loop: Where all other parameters of the model are optimized, such as linear projections, MLP layers, convolutions, etc.

## D.1 EXAMPLES

As an example, one can define the linear attention as the optimization of dot-product similarity with gradient descent: i.e., $\tilde{\ell}_t := \langle \mathcal{M}_{t-1} \boldsymbol{k}_t, \mathbf{v}_t \rangle$. That is,

$$\mathcal{M}_t = \mathcal{M}_{t-1} - \eta_t \nabla \tilde{\ell}_t(\mathcal{M}_{t-1}; \boldsymbol{k}_t, \mathbf{v}_t) = \mathcal{M}_{t-1} - \eta_t \nabla \langle \mathcal{M}_{t-1} \boldsymbol{k}_t, \mathbf{v}_t \rangle \quad (18)$$

$$= \mathcal{M}_{t-1} + \eta_t \mathbf{v}_t \boldsymbol{k}_t^\top. \quad (19)$$

As an another example, if we use regression loss, instead of the dot-product similarity, we can obtain the DeltaNet (Schlag et al., 2021):

$$\mathcal{M}_t = \mathcal{M}_{t-1} - \eta_t \nabla \|\mathcal{M}_t \boldsymbol{k}_t - \mathbf{v}_t\|_2^2 = \mathbf{I} - \eta_t \boldsymbol{k}_t \boldsymbol{k}_t^\top \mathcal{M}_{t-1} + \mathbf{v}_t \boldsymbol{k}_t^\top. \quad (20)$$

# E SUPPORTING PROOFS

**Proposition 1** (Capacity of $\ell_2$ Attentional Bias). *Let $\mathcal{M}$ be a matrix-valued memory with $d_v \times d_k$ parameters that optimizes the internal objective of $\ell(\mathcal{M}_t; \boldsymbol{k}_t, \mathbf{v}_t) = \|\mathcal{M}_t \boldsymbol{k}_t - \mathbf{v}_t\|_2^2$ with gradient descent. $\mathcal{M}$ can store the mapping of at most $\mathcal{O}(d_k)$ pairs of $(\boldsymbol{k}_i, \mathbf{v}_i)$ with linearly independent keys.*

*Proof.* Let $K = [\boldsymbol{k}_1 \cdots \boldsymbol{k}_m] \in \mathbb{R}^{d_k \times m}$ and $V = [\mathbf{v}_1 \cdots \mathbf{v}_m] \in \mathbb{R}^{d_v \times m}$. The optimization problem becomes minimizing the Frobenius norm $\|\mathcal{M}K - V\|_2^2$. Exact memorization requires solving the linear system $\mathcal{M}K = V$.

Vectorizing the expression yields the system $(K^\top \otimes I_{d_v}) \text{vec}(\mathcal{M}) = \text{vec}(V)$, which has $md_v$ scalar equations in $d_k d_v$ unknowns. When the keys are linearly independent, $\text{rank}(K) = m$, and hence the system matrix has full row rank $md_v$. Solvability thus requires $md_v \leq d_k d_v$, or equivalently $m \leq d_k$. This matches classic results on the storage capacity of linear associative memories such as the Willshaw model and Hopfield networks, where capacity is tied to the rank of the input embedding (Willshaw et al., 1969; Hopfield, 1982).

When $m \leq d_k$ and $K$ has full column rank, one can construct an exact interpolating solution via the Moore–Penrose pseudoinverse: $\mathcal{M}^* = VK^\top$. Then $\mathcal{M}^*K = VK^\top K = V$, achieving zero training error. Thus the upper bound is tight.

Moreover, full-batch gradient descent on this objective with step size $0 < \eta < 2/\lambda_{\max}(KK^\top)$ yields iterates $\mathcal{M}_{t+1} = \mathcal{M}_t - \eta(\mathcal{M}_t K - V)K^\top$, which converge to the minimum-norm interpolating solution $\mathcal{M}^\dagger = VK^\top$ when $m \leq d_k$. This is a well-known implicit bias of gradient descent in overparameterized linear models (Satpathi & Srikant, 2021).

Finally, the same rank-based constraint governs the capacity of linear or multi-head attention modules. In such architectures, the output context matrix has rank at most $\text{rank}(K) \leq d_k$, which directly limits their expressivity. Recent analyses identify this "low-rank bottleneck" as a capacity-limiting effect in Transformers (Bhojanapalli et al., 2020). □

**Theorem 1** (Effect of Deep Memory). *Let $\mathcal{M}(\cdot)$ be an MLP with $\mathcal{L}_{\mathcal{M}} \geq 2$ layers, $d_k$ input dimension, and $d_h$ hidden dimension. Then, $\mathcal{M}(\cdot)$ can store the mapping of at least $\mathcal{O}(d_k d_v)$ and at most $\mathcal{O}\left(d_k d_v \sum_{i=1}^{\mathcal{L}_{\mathcal{M}}} \min\{d_h^{(j)}\}_{j \geq i} d_h^{(j+1)}\right)$ pairs of $(\boldsymbol{k}_i, \mathbf{v}_i)$ with linearly independent keys.*

Early theoretical works established that even simple network architectures can memorize a significant number of input-output mappings, with capacity often related to the number of network

parameters (e.g., weights and biases) and the input dimensionality Cover (1965); Baum (1988); Huang (2003). For instance, Baum (1988) demonstrated that $\left\lceil \frac{N}{d} \right\rceil$ neurons are sufficient for a single-hidden-layer network with threshold units to memorize $N$ input-label pairs from $\mathbb{R}^d$.

Networks employing Rectified Linear Units (ReLUs), exhibit a piecewise affine behavior. The input space is partitioned into numerous linear regions, and within each region, the network computes a distinct affine transformation Montufar et al. (2014); Pascanu et al. (2014). This structure is pivotal for analyzing their expressive power and storage capacity. The precise relationship between depth, width, the number of linear regions, and the ultimate capacity to store specific key-value associations, especially with constraints like linearly independent keys, remains an active area of research.

*Proof.* Let $m$ denote the number of $(\boldsymbol{k}_i, \mathbf{v}_i)$ pairs memorized exactly by $\mathcal{M}$, and assume the keys $\{\boldsymbol{k}_i\}_{i=1}^m \subset \mathbb{R}^{d_k}$ are linearly independent. Let $d_h^{(0)} := d_k$, $d_h^{(\mathcal{L_M})} := d_v$, and for each layer $1 \leq \ell \leq \mathcal{L_M}$, define $W^{(\ell)} \in \mathbb{R}^{d_h^{(\ell)} \times d_h^{(\ell-1)}}$. Biases are omitted for simplicity.

Since $\sigma(x) = \max(0, x)$ is piecewise linear, the composition of linear maps and ReLU activations yields a piecewise affine function. For any fixed activation pattern (i.e., fixed sign of pre-activations), the MLP acts as:

$$\mathcal{M}(\cdot) = A \cdot + B, \quad \text{where } A = W^{(\mathcal{L_M})} D^{(\mathcal{L_M}-1)} W^{(\mathcal{L_M}-1)} \cdots D^{(1)} W^{(1)},$$

and each $D^{(\ell)}$ is a diagonal $\{0,1\}$ matrix selecting the active units. Therefore, when all keys fall into the same linear region (which occurs generically after a small perturbation), $\mathcal{M}$ is a single affine transformation.

Let $\mathbf{K} := [\boldsymbol{k}_1 \ \cdots \ \boldsymbol{k}_m] \in \mathbb{R}^{d_k \times m}$ and $\mathbf{V} := [\mathbf{v}_1 \ \cdots \ \mathbf{v}_m] \in \mathbb{R}^{d_v \times m}$. Exact memorization implies $A\mathbf{K} = \mathbf{V}$, so:

$$\text{rank}(\mathbf{V}) \leq \text{rank}(A), \quad m = \text{rank}(\mathbf{K}) \leq \min\{\text{rank}(A), d_k\}.$$

Now observe:

$$A = W^{(\mathcal{L_M})} \underbrace{D^{(\mathcal{L_M}-1)} W^{(\mathcal{L_M}-1)}}_{R_{\mathcal{L_M}-1}} \cdots \underbrace{D^{(1)} W^{(1)}}_{R_1},$$

and thus the rank of $A$ is bounded by the minimal width encountered along each path times the immediate input dimension:

$$\text{rank}(A) \leq \sum_{i=1}^{\mathcal{L_M}} \left( \min_{j \geq i} d_h^{(j)} \right) d_h^{(i)} = \mathcal{O}\left( d_k d_v \sum_{i=1}^{\mathcal{L_M}} \min_{j \geq i} d_h^{(j)} d_h^{(i+1)} \right).$$

Hence,

$$m \leq \mathcal{O}\left( d_k d_v \sum_{i=1}^{\mathcal{L_M}} \min_{j \geq i} d_h^{(j)} d_h^{(i+1)} \right)$$

$\square$

**Proposition 2** (Memory Capacity with Polynomial Mapping)**.** *Let $\phi_p(\cdot)$ be a polynomial mapping with degree at most $p$, and $\mathcal{M}$ be a matrix-valued memory that optimizes the internal objective of $\ell(\mathcal{M}_t; \phi_p(\boldsymbol{k}_t), \mathbf{v}_t) = \|\mathcal{M}_t \phi_p(\boldsymbol{k}_t) - \mathbf{v}_t\|_2^2$ with gradient descent. $\mathcal{M}$ can store the mapping of at most $\mathcal{O}\left(d_k{}^p\right)$ pairs of $(\boldsymbol{k}_i, \mathbf{v}_i)$ with linearly independent keys, where $d_k$ is the dimension of keys $\boldsymbol{k}_i$.*

*Proof.* Let us begin by analyzing the dimension of the lifted feature space induced by $\phi_p$. A monomial in $d_k$ variables of total degree exactly $\ell$ has the form $\boldsymbol{k}^\alpha = \prod_{j=1}^{d_k} k_j^{\alpha_j}$, where $\alpha \in \mathbb{N}^{d_k}$ and

$|\alpha| := \sum_{j=1}^{d_k} \alpha_j = \ell$. The number of such monomials is given by the classical stars-and-bars formula, which counts the number of integer solutions to $\alpha_1 + \cdots + \alpha_{d_k} = \ell$, yielding

$$\binom{d_k + \ell - 1}{\ell}.$$

Summing over all degrees $\ell = 0$ to $p$ gives the total number of monomials (i.e., the output dimension of $\phi_p$),

$$D = \sum_{\ell=0}^{p} \binom{d_k + \ell - 1}{\ell} = \binom{d_k + p}{p},$$

where the final identity follows from the hockey-stick identity in combinatorics.

To characterize the memorization capacity, we reformulate the loss in matrix notation. Let $\Phi := [\phi_p(\boldsymbol{k}_1) \ \cdots \ \phi_p(\boldsymbol{k}_m)] \in \mathbb{R}^{D \times m}$ and $V := [\mathbf{v}_1 \ \cdots \ \mathbf{v}_m] \in \mathbb{R}^{d_v \times m}$. Then the objective becomes

$$L(\mathcal{M}) = \tfrac{1}{2}\|\mathcal{M}\Phi - V\|_2^2.$$

Exact memorization corresponds to the existence of a matrix $\mathcal{M}$ such that $\mathcal{M}\Phi = V$. This is a linear system in which $\mathcal{M}$ acts on the columns of $\Phi$, so the rank of $\Phi$ necessarily limits the number of independent targets $\mathbf{v}_i$ that can be fitted exactly.

By the sub-multiplicativity of rank, for any matrices $A$ and $B$, we have

$$\mathrm{rank}(AB) \leq \min\{\mathrm{rank}(A), \mathrm{rank}(B)\}.$$

Applying this to $\mathcal{M}\Phi$ yields

$$\mathrm{rank}(\mathcal{M}\Phi) \leq \mathrm{rank}(\Phi) \leq D.$$

Now consider a case where the targets $\mathbf{v}_1, \ldots, \mathbf{v}_m$ are linearly independent; for instance, take $V = [e_1, \ldots, e_m]$, the first $m$ standard basis vectors. Then $\mathrm{rank}(V) = m$. If $m > D$, we necessarily have $\mathrm{rank}(\mathcal{M}\Phi) < \mathrm{rank}(V)$ for every choice of $\mathcal{M}$, implying that the system $\mathcal{M}\Phi = V$ is unsolvable. Hence, the loss remains strictly positive, and exact memorization is impossible.

This establishes that no method, regardless of optimization procedure, can memorize more than $D = \binom{d_k + p}{p}$ independent input-output pairs under a degree-$\leq p$ polynomial lifting. Since $\binom{d_k + p}{p} = \Theta(d_k^p)$ for fixed $p$, the result follows: the memorization capacity is bounded above by $\mathcal{O}(d_k^p)$. $\qquad\square$

## F  DETAILED FORMULATIONS OF ALL ARCHITECTURES

In this section, for the sake of clarity, we discuss the details of all architectures that we discuss through the paper:

### F.1  DEEP LINEAR ATTENTION (DLA)

We design Deep Linear Attention (DLA)—linear attention module that uses a deep MLP as the memory (KV cache)—as one of the baselines of this study. Given input $\boldsymbol{x} \in \mathbb{R}^{N \times d_{\mathrm{in}}}$, we project the input into matrices of keys, values and queries:

$$\mathbf{Q} = \begin{pmatrix} \boldsymbol{q}_1 \\ \vdots \\ \boldsymbol{q}_N \end{pmatrix} = \boldsymbol{x}\mathbf{W}_Q, \qquad \mathbf{K} = \begin{pmatrix} \boldsymbol{k}_1 \\ \vdots \\ \boldsymbol{k}_N \end{pmatrix} = \boldsymbol{x}\mathbf{W}_K, \qquad \mathbf{V} = \begin{pmatrix} \mathbf{v}_1 \\ \vdots \\ \mathbf{v}_N \end{pmatrix} = \boldsymbol{x}\mathbf{W}_V, \quad (21)$$

where $\mathbf{W}_Q, \mathbf{W}_K$, and $\mathbf{W}_V$ are learnable linear layers. We then define memory as a learning module that optimizes the inner-dot product similarity using gradient descent: i.e.,

$$\min_{\mathcal{M}} \underbrace{\langle \mathcal{M}(\boldsymbol{k}_t), \mathbf{v}_t \rangle}_{\ell(\mathcal{M}_{t-1}; \boldsymbol{k}_t, \mathbf{v}_t)}. \qquad (22)$$

The above optimization using gradient descent results in the following recurrence (we also add weight decay with input-dependent parameter $\alpha_t$):

$$\mathcal{M}_t = \alpha_t \mathcal{M}_{t-1} - \eta_t \nabla \ell(\mathcal{M}_{t-1}; \boldsymbol{k}_t, \mathbf{v}_t) \tag{23}$$

which in the case of linear memory (i.e., $\mathcal{M}_t = W_t \in \mathbb{R}^{d \times d}$) it becomes:

$$W_t = \alpha_t W_{t-1} + \mathbf{v}_t \boldsymbol{k}_t^\top, \tag{24}$$

which is the formulation of gated linear attention. We use the same training process as other models (see Appendix H).

## F.2 SLIDING WINDOW LINEAR ATTENTION (SWLA)

The design of SWLA is the same as the design of DLA, but with the use of sliding window objective. That is, given keys, values, and queries:

$$\mathbf{Q} = \begin{pmatrix} \boldsymbol{q}_1 \\ \vdots \\ \boldsymbol{q}_N \end{pmatrix} = \boldsymbol{x} \mathbf{W}_Q, \qquad \mathbf{K} = \begin{pmatrix} \boldsymbol{k}_1 \\ \vdots \\ \boldsymbol{k}_N \end{pmatrix} = \boldsymbol{x} \mathbf{W}_K, \qquad \mathbf{V} = \begin{pmatrix} \mathbf{v}_1 \\ \vdots \\ \mathbf{v}_N \end{pmatrix} = \boldsymbol{x} \mathbf{W}_V, \tag{25}$$

we optimize the internal objective of:

$$\min_{\mathcal{M}} \underbrace{\sum_{i=t-c+1}^{t} \langle \mathcal{M}_{t-1}(\boldsymbol{k}_i), \mathbf{v}_i \rangle}_{\ell(\mathcal{M}_{t-1}; \boldsymbol{k}_t, \mathbf{v}_t)}. \tag{26}$$

The above formulation, results in:

$$\mathcal{M}_t = \alpha_t \mathcal{M}_{t-1} - \nabla \ell(\mathcal{M}_{t-1}; \boldsymbol{k}_t, \mathbf{v}_t) = \alpha_t \mathcal{M}_{t-1} - \sum_{i=t-c+1}^{t} \eta_i^{(t)} \nabla \langle \mathcal{M}_{t-1}(\boldsymbol{k}_i), \mathbf{v}_i \rangle, \tag{27}$$

which in the case of linear memory (i.e., $\mathcal{M}_t = W_t \in \mathbb{R}^{d \times d}$) it becomes:

$$\mathcal{M}_t = \alpha_t \mathcal{M}_{t-1} - \sum_{i=t-c+1}^{t} \eta_i^{(t)} \mathbf{v}_i \boldsymbol{k}_i^\top. \tag{28}$$

## F.3 OMEGANET

In the design of OMEGANET, we use replace the dot-prodcut similarity objective with $\ell(\mathcal{M}_{t-1}; \boldsymbol{k}_t, \mathbf{v}_t) = \sum_{i=t-c+1}^{t} \|\mathcal{M}_{t-1}(\phi(\boldsymbol{k}_i)) - \mathbf{v}_i\|_2^2$ ,which results in the recurrence of:

$$\mathcal{M}_t = \alpha_t \mathcal{M}_{t-1} - \nabla \ell(\mathcal{M}_{t-1}; \boldsymbol{k}_t, \mathbf{v}_t) = \alpha_t \mathcal{M}_{t-1} - \sum_{i=t-c+1}^{t} \eta_i^{(t)} \nabla \|\mathcal{M}_{t-1}(\phi(\boldsymbol{k}_i)) - \mathbf{v}_i\|_2^2. \tag{29}$$

In the above formulation, $\phi(.)$ is the polynomial feature mapping function.

## F.4 ATLAS

In the ATLAS, we use the same internal objective as OMEGANET but we optimize it using Muon optimizer (Jordan et al., 2024) with weight decay. That is,

$$\mathcal{M}_t = \alpha_t \mathcal{M}_{t-1} + \texttt{Newton-schulz5}(\mathcal{S}_t) \tag{30}$$

$$\mathcal{S}_t = \theta_t \mathcal{S}_{t-1} - \sum_{i=t-c+1}^{t} \eta_i^{(t)} \nabla \|\mathcal{M}_{t-1}(\phi(\boldsymbol{k}_i)) - \mathbf{v}_i\|_2^2. \tag{31}$$

## G  ADDITIONAL DISCUSSIONS ON THE OMEGA RULE'S VARIANTS

**Beyond Gradient Descent.** The concept of Omega rule and "test time memorization of context" can simply be extended to optimizing the objective in Equation 6 with any arbitrary optimizer, even beyond simple gradient descent. We use two extreme cases for $c$ as the illustrations. In the first case, we let $c = 1$, $\gamma_i^{(t)} = 1$, and use gradient descent with momentum as the optimizers, resulting in the following update rule:

$$\mathcal{M}_t = \alpha_t \mathcal{M}_{t-1} + \mathcal{S}_t \tag{32}$$

$$\mathcal{S}_t = \theta_t \mathcal{S}_{t-1} - \eta_t \nabla \ell(\mathcal{M}_{t-1}; \boldsymbol{k}_t, \mathbf{v}_t). \tag{33}$$

This update rule is equivalent to the long-term neural memory in Titans (Behrouz et al., 2024). In the second case, using a linear memory $\mathcal{M}$, letting $\gamma_i^{(t)} = 1$, and $c$ be equal to the context length, the memory update process is equivalent to optimizing the (regularized) least-squares problem:

$$\mathcal{M}_t = \min_{\mathcal{M}} \sum_{i=1}^{t} \|\mathcal{M} \boldsymbol{k}_i - \mathbf{v}_i\|_2^2. \tag{34}$$

Von Oswald et al. (2023) suggest directly optimizing the above objective and use Sherman-Morrison formula (Sherman & Morrison, 1950) to recursively calculate the inverse term in the optimal solution. Despite the optimality of memory, such direct solutions comes with the cost of non-parallelizable training and also are limited to only the linear matrix-valued memory setup. Furthermore, as discussed earlier, the global nature without any direct hard gating terms (i.e., $\gamma_i^{(t)}$s) can force the model to *not* prune the context, damaging the performance in longer sequences.

### G.1  MEMORY CAPACITY AND EXPONENTIAL KERNELS

We first recall the formulation of `softmax` attention in Transformers (i.e., Equation 15):

$$\mathbf{y}_i = \frac{1}{\sum_{\ell=1}^{i} \exp\left(\mathbf{q}_i^\top \mathbf{k}_\ell / \sqrt{d_{\text{in}}}\right)} \sum_{j=1}^{i} \exp\left(\mathbf{q}_i^\top \mathbf{k}_j / \sqrt{d_{\text{in}}}\right) \mathbf{v}_j, \tag{35}$$

which its $\exp(\cdot)$ kernel is not separable and so cannot be written as a recurrence. Following the discussion in Kacham et al. (2024b), one can see $\exp(\cdot)$ kernel (compared to polynomial kernel $\phi_p(\cdot)$) as a feature map that maps the input into an infinite dimension. That is, we define:

$$\phi^*(x) = \begin{pmatrix} 1 \\ \frac{x}{\sqrt{1}} \\ \frac{x^{\otimes 2}}{\sqrt{2!}} \\ \frac{x^{\otimes 3}}{\sqrt{3!}} \\ \vdots \end{pmatrix}, \qquad \phi_p(x) = x^{\otimes p}, \tag{36}$$

where $x^{\otimes p} = x \otimes x^{\otimes(p-1)}$ is a "self-tensoring" operator with Kronecker product (Kacham et al., 2024b) and so:

$$\exp(\boldsymbol{q}_t^\top \boldsymbol{k}_t) = \phi^*(\boldsymbol{q}_t)^\top \phi^*(\boldsymbol{k}_t). \tag{37}$$

Based on the above kernel, we can reformulate the attention (see Equation 35) as: (we remove $1/\sqrt{d_{\text{in}}}$ term for the sake of simplicity)

$$\mathbf{y}_i = \frac{1}{\sum_{\ell=1}^{i} \exp\left(\mathbf{q}_i^\top \mathbf{k}_\ell / \sqrt{d_{\text{in}}}\right)} \sum_{j=1}^{i} \mathbf{v}_j \phi^*(\mathbf{k}_j)^\top \phi^*(\mathbf{q}_i) \tag{38}$$

$$= \frac{1}{\sum_{\ell=1}^{i} \exp\left(\mathbf{q}_i^\top \mathbf{k}_\ell / \sqrt{d_{\text{in}}}\right)} \left(\sum_{j=1}^{i} \phi^*(\mathbf{v}_j \mathbf{k}_j)^\top\right) \phi^*(\mathbf{q}_i) = \mathcal{M}_i \phi^*(\mathbf{q}_i), \tag{39}$$

This formulation, provides another important insight on the differences of attention and (kernel) recurrent models: `Softmax` attention as an associative memory has an unbounded memory and so can better memorize larger context into its parameters. Building upon this insight, we present DEEPTRANSFORMERS by replacing polynomial kernel with $\phi^*(\cdot)$ kernel in Deep Linear Attention formulation (Equation 11), resulting in unnormalized formulation of:

$$\mathcal{M}_t = \mathcal{M}_{t-1} - \nabla\langle\mathcal{M}_{t-1}(\phi^*(\boldsymbol{k}_t)), \mathbf{v}_t\rangle. \tag{40}$$

In the special case of linear memory, we can derive the closed form for the above formulation as:

$$\mathcal{M}_t = \mathcal{M}_{t-1} - \nabla\langle\mathcal{M}_{t-1}\phi^*(\boldsymbol{k}_t), \mathbf{v}_t\rangle = \mathcal{M}_{t-1} + \mathbf{v}_t\phi^*(\boldsymbol{k}_t)^\top = \sum_{i=1}^t \mathbf{v}_i\phi^*(\boldsymbol{k}_i)^\top \tag{41}$$

$$\Rightarrow \quad \mathbf{y}_t = \mathcal{M}_t\phi^*(\boldsymbol{q}_t) = \sum_{i=1}^t \mathbf{v}_i\exp(\boldsymbol{q}_i^\top\boldsymbol{k}_i), \tag{42}$$

which matches the output of the unnormalized Transformers. Therefore, DEEPTRANSFORMERS are strict generalizations of Transformers with `softmax` attention (Vaswani et al., 2017).

### G.2 DEEP OMEGA TRANSFORMER (DOT): TRANSFORMERS WITH OMEGA LEARNING RULE

Our above formulation of DEEPTRANSFORMERS is based on the (Hebbian Rule), which is also used in original Transformers. However, as discussed earlier, using more powerful memory management and learning rules in associative memory modules can further enhance their performance. To this end, we extend the above formulation by replacing the Hebbian rule with our Omega learning rule, resulting in an unnormalized formulation of Deep Omega Transformers (DOT):

$$\mathcal{M}_t = \mathcal{M}_{t-1} - \nabla\sum_{i=t-c+1}^t \gamma_i^{(t)}\|\mathcal{M}(\phi^*(\boldsymbol{k}_i)) - \mathbf{v}_i\|_2^2. \tag{43}$$

We now discuss special instances of DOT to provide further intuition on its generalized formulation.

**Linear Memory.** This setup results in the following unnormalized formulation:

$$\mathcal{M}_t = \left(\mathbf{I} - \sum_{i=t-c+1}^t \gamma_i^{(t)}\phi^*(\boldsymbol{k}_i)\phi^*(\boldsymbol{k}_i)^\top\right)\mathcal{M}_{t-1} - \sum_{i=t-c+1}^t \gamma_i^{(t)}\mathbf{v}_i\phi^*(\boldsymbol{k}_i)^\top \tag{44}$$

$$\Rightarrow \mathbf{y}_t = \mathcal{M}_t\phi^*(\boldsymbol{q}_t) = \left(\mathbf{I} - \sum_{i=t-c+1}^t \gamma_i^{(t)}\phi^*(\boldsymbol{k}_i)\phi^*(\boldsymbol{k}_i)^\top\right)\mathcal{M}_{t-1}\phi^*(\boldsymbol{q}_t) - \sum_{i=t-c+1}^t \gamma_i^{(t)}\mathbf{v}_i\exp(\boldsymbol{q}_t^\top\boldsymbol{k}_i). \tag{45}$$

**Online Case with $c = 1$.** We now let $c = 1$:

$$\mathcal{M}_t = \left(\mathbf{I} - \eta_t\phi^*(\boldsymbol{k}_t)\phi^*(\boldsymbol{k}_t)^\top\right)\mathcal{M}_{t-1} - \eta_t\mathbf{v}_t\phi^*(\boldsymbol{k}_t)^\top \tag{46}$$

$$\Rightarrow \mathbf{y}_t = \mathcal{M}_t\phi^*(\boldsymbol{q}_t) = \left(\mathbf{I} - \eta_t\phi^*(\boldsymbol{k}_t)\exp(\boldsymbol{q}_t^\top\boldsymbol{k}_t)\right)\mathcal{M}_{t-1} - \eta_t\mathbf{v}_t\exp(\boldsymbol{q}_t^\top\boldsymbol{k}_t). \tag{47}$$

The above (unnormalized) formulation can be seen as the generalization of Transformers with Delta Rule. Therefore, due to the unbounded memory, DOT not only appends the new keys and values (similar to original Transformers), but it also replaces the new value with its predicted value from the previous state.

## H PARALLELIZING OMEGA RULE

While Omega rule provides a more general and expressive formulation for the design of memory modules than Hebbian or Delta learning rules, its applicability to large-scale models relies on its efficiency in training. To this end, we discuss a fast parallelizable training algorithms that does not add any significant computational overhead with the online counterpart version (i.e., $c = 1$). A naive implementation requires materializing $c$ gradients $\nabla\ell \in \mathbb{R}^{d_{\text{in}} \times d_{\text{in}}}$, which can result in a significantly

higher memory footprint and I/O cost when $d_{\text{in}}$ is large. Also, to fully utilize hardware accelerators such as TPUs and GPUs, it is important to tensorize computations and maximize the use of `matmul` operations. Motivated by recent work (Behrouz et al., 2024; Sun et al., 2024), we propose a simple sliding window masking strategy that supports efficient parallel training while avoiding substantial memory overhead. Specifically, we partition the input sequence with length $L$ into chunks of size $b \geq 1$, each of which is represented by $\mathbf{S}_i = \{\boldsymbol{x}_{(i-1)b+1}, \ldots, \boldsymbol{x}_{ib}\}$. Then for each chunk, we calculate the gradients with respect to the last state of the previous chunk. For the sake of clarity, we first assume $\gamma_i^{(t)} = \eta_t$ for all positions in the sequence. When the chunk size is $b = 1$, the update rule is:

$$\mathcal{M}_t = \alpha_t \mathcal{M}_{t-1} - \eta_t \sum_{i=t-c+1}^{t} \nabla\ell(\mathcal{M}_{t-1}; \boldsymbol{k}_i, \mathbf{v}_i), \tag{48}$$

where $\mathcal{M}_t$ is the model state at step $t$, $\alpha_t$ and $\eta_t$ are the weight decay and learning rate parameters respectively, and $(\boldsymbol{k}_i, \mathbf{v}_i)$ denote the input pair at position $i$. In practice, we strike a balance between the fully recurrent form and the fully parallel form by dividing the sequence into smaller chunks. Within each chunk (intra-chunk), we apply parallel computation, while across chunks (inter-chunk), we adopt a recurrent computation scheme. We now define $t' = t - \text{mod}(t, b)$. That is, for time steps $t$ such that $t' \leq t < t' + b$, the update rule within each chunk becomes:

$$\mathcal{M}_t = \alpha_t \ldots \alpha_{t'} \mathcal{M}_{t'} - \sum_{n=t'}^{t} \frac{\alpha_t \ldots \alpha_{t'}}{\alpha_n \ldots \alpha_{t'}} \eta_n \underbrace{\sum_{i=n-c+1}^{n} \nabla\ell(\mathcal{M}_{t'}; \boldsymbol{k}_i, \mathbf{v}_i)}_{G_t} \tag{49}$$

In our implementation, for $G_t$, we follow the same gradient computation approach as described in Titans (Behrouz et al., 2024) but additionally apply a sliding window mask $M_s$ during the broadcasting operation (e.g., using `einsum`). When $c = 1$, the sliding window mask $M_s$ reduces to the identity matrix. For $c > 1$, $M_s$ is an identity matrix except that the $c - 1$ positions immediately preceding each diagonal entry are also set to 1. This allows gradient contributions from a window of size $c$, enabling efficient computation without materializing all gradients inside the chunk.

## H.1 PARALLEL TRAINING

In this section, we discussed how the training process of ATLAS can be parallelized. For the sake of clarity, we assume $c = 1$. Generalizing the process to arbitrary value for $c$ follows the procedure in Appendix H. We use the same process as we discussed in Appendix H and so chunk the sequence and compute all the gradients with respect to the last state of the previous chunk. Accordingly, using the recurrence of ATLAS with momentum but without , we have:

$$\mathcal{M}_t = \alpha_t \mathcal{M}_{t-1} + \mathcal{S}_t \tag{50}$$
$$\mathcal{S}_t = \theta_t \mathcal{S}_{t-1} - \eta_t \nabla\ell(\mathcal{M}_{t'}; \boldsymbol{k}_t, \mathbf{v}_t). \tag{51}$$

Since $t'$ is the last state of the previous chunk, we can calculate all the gradients before hand and so we let $u_t = \nabla\ell(\mathcal{M}_{t'}; \boldsymbol{k}_t, \mathbf{v}_t)$. Therefore, we have:

$$\mathcal{M}_t = \alpha_t \mathcal{M}_{t-1} + \mathcal{S}_t \tag{52}$$
$$\mathcal{S}_t = \theta_t \mathcal{S}_{t-1} - \eta_t u_t. \tag{53}$$

Now by expanding the second recurrence, we have:

$$\mathcal{S}_t = \theta_t \mathcal{S}_{t-1} - \eta_t \underbrace{\nabla\ell(\mathcal{M}_{t'}; \boldsymbol{k}_t, \mathbf{v}_t)}_{u_t}, \tag{54}$$

$$\Rightarrow \mathcal{S}_t = \underbrace{\theta_t \ldots \theta_1}_{\beta_t} \mathcal{S}_0 - \sum_{i=1}^{t} \frac{\theta_t \ldots \theta_1}{\theta_i \ldots \theta_1} \eta_i u_i = \beta_t \mathcal{S}_0 - \Theta \odot E \odot G, \tag{55}$$

where $G$ is the gradient matrix, $E$ and $\Theta$ are diagonal matrices with value $\theta$ and $\eta$, and $\odot$ is broadcasting.

Table 4: Architectural Details.

| Model | Block | Dim | Head | Peak LR | Token |
|-------|-------|------|------|---------|-------|
| 170M | 12 | 768 | 16 | 3e-3 | 15B |
| 340M | 24 | 1024 | 16 | 1.5e-3 | 15B |
| 760M | 24 | 1536 | 16 | 1.25e-3 | 30B |
| 1.3B | 18 | 2048 | 8 | 7e-4 | 100B |

The main advantage of the above formulation (chunk wise recurrence) is that the recurrence of momentum is independent of the state of memory. That is, we can calculate all the momentum terms in the beginning of the chunk using the above formulation. Now in the Muon case, we want to use Newton-Schulz algorithm on the momentum terms, which results in:

$$\mathcal{S}'_t \leftarrow \texttt{Newton-Schulz5}(\mathcal{S}_t), \tag{56}$$

$$\mathcal{M}_t = \mathcal{M}_{t-1} + \mathcal{S}'_t. \tag{57}$$

Since the calculation of all $\mathcal{S}_t$s can be done in parallel, the calculation of $\texttt{Newton-Schulz5}(\cdot)$ can also be done in parallel.

## I    EXPERIMENTAL DETAILS

In our experimental setup we follow recent studies on linear recurrent models (Yang et al., 2024a; Behrouz et al., 2024; 2025), we use Wikitext (Merity et al., 2017), LMB (Paperno et al., 2016), PIQA (Bisk et al., 2020), HellaSwag (Zellers et al., 2019), WinoGrande (Sakaguchi et al., 2021), ARC-easy (ARC-e) and ARC-challenge (ARC-c) (Clark et al., 2018), SIQA (Sap et al., 2019), and BoolQ (Clark et al., 2019). Also, the baselines results are from Behrouz et al. (2025; 2024). In the training, we use a vocabulary size of 32K and use training length of 4K tokens (2K for SWA). We employ AdamW optimizer with learning rate of $4e$-$4$ with cosine annealing schedule with batch size of 0.5M tokens, and weight decay of $0.1$. The architectural details are also reported in Table 4. The baseline results for 1.3B are from Yang et al. (2024a) and for 760M are from Behrouz et al. (2024; 2025).

For the memory architecture, unless state otherwise, we use an MLP with 2 layers with expansion factor of 4 and GELU activation function (Hendrycks & Gimpel, 2016). We also use residual connections and layer norm at the end of each chunk: $\mathcal{M}(x) = x + W_1\sigma(W_2 x)$.

## J    ADDITIONAL EXPERIMENTAL RESULTS

In this section, we provide additional experimental results to support the design of our models, understand the effect of different components and also evaluate their performance in long context, in-context recall and MAD tasks.

### J.1    LANGUAGE MODELING AND COMMON-SENSE REASONING (FULL RESULTS)

In Section 5 we presented a subset of results on language modeling and common-sense reasoning tasks. In this section, we further report the results for all scales of models. The results are in Table 5.

**State-of-the-art Results.** Looking at the performance of ATLAS and OMEGANET, both architectures perform favorably compared to modern linear recurrent models and Transformers, achieving lower perplexity and better accuracy in downstream tasks. Even the fully recurrent version of these models outperform hybrid models such as Samba (Ren et al., 2024) and Gated DeltaNet-H2 (Yang et al., 2024a). Using the hybrid variants of MAG and MAL further improve the performance of ATLAS, which shows the complementary role of recurrent long-term memory and attention.

**The Effect of Design.** Comparing the performance of ATLAS, OMEGANET, and baselines SWLA and DLA, we can see the role of $\ell_2$ regression loss as the attentional bias. Also, the better performance of SWLA compared to GLA and RetNet indicates the importance of memorizing the context, instead of memorizing individual tokens.

Table 5: Performance of ATLAS and baselines on language modeling and common-sense reasoning tasks. Hybrid models are marked with *. The best results are highlighted highlighted .

| Model | Wiki. ppl↓ | LMB. ppl↓ | LMB. acc↑ | PIQA acc↑ | Hella. acc_n↑ | Wino. acc↑ | ARC-e acc↑ | ARC-c acc_n↑ | SIQA acc↑ | BoolQ acc↑ | Avg. ↑ |
|---|---|---|---|---|---|---|---|---|---|---|---|
| | | | | 340M params / 15B tokens | | | | | | | |
| Transformer++ | 31.52 | 41.08 | 30.76 | 62.98 | 34.76 | 50.53 | 45.21 | 24.05 | 36.81 | 58.24 | 42.92 |
| RetNet | 32.50 | 49.73 | 28.24 | 62.61 | 34.15 | 50.91 | 44.27 | 23.62 | 36.79 | 59.72 | 42.54 |
| GLA | 28.51 | 43.02 | 28.73 | 64.05 | 35.96 | 50.00 | 54.19 | 24.29 | 37.13 | 58.39 | 44.09 |
| Mamba | 30.83 | 40.21 | 29.94 | 63.79 | 35.88 | 49.82 | 49.24 | 24.56 | 35.41 | 60.07 | 43.59 |
| DeltaNet | 28.65 | 47.30 | 28.43 | 63.52 | 35.95 | 49.63 | 52.68 | 25.37 | 37.96 | 58.79 | 44.04 |
| TTT | 27.44 | 34.19 | 30.06 | 63.97 | 35.71 | 50.08 | 53.01 | 26.11 | 37.32 | 59.83 | 44.51 |
| Gated DeltaNet | 27.01 | 30.94 | 34.11 | 63.08 | 38.12 | 51.60 | 55.28 | 26.77 | 34.89 | 59.54 | 45.42 |
| MONETA | 26.19 | 29.31 | 35.70 | 63.99 | 39.23 | 52.04 | 55.96 | 27.15 | 37.29 | 60.22 | 46.44 |
| YAAD | 26.61 | 29.11 | 34.09 | 64.93 | 39.86 | 51.12 | 54.75 | 28.64 | 33.82 | 60.29 | 45.93 |
| MEMORA | 27.16 | 30.44 | 33.68 | 65.21 | 39.17 | 51.23 | 53.40 | 27.99 | 34.1 | 59.29 | 45.51 |
| DLA (ours) | 27.93 | 35.09 | 30.8 | 62.9 | 36.2 | 50.4 | 53.5 | 26.7 | 37.1 | 59.7 | 44.76 |
| SWDT (ours) | 26.98 | 33.95 | 32.4 | 63.1 | 38.2 | 50.9 | 54.9 | 27.5 | 37.5 | 59.6 | 45.31 |
| OMEGANET (ours) | 26.03 | 28.76 | 35.6 | 65.3 | 39.7 | 52.0 | 56.1 | 28.6 | 37.7 | 60.4 | 46.93 |
| ATLAS (ours) | 25.88 | 28.54 | 36.1 | 64.9 | 40.1 | 52.7 | 56.4 | 28.8 | 38.1 | 61.2 | 47.28 |
| | | | | 760M params / 30B tokens | | | | | | | |
| Transformer++ | 25.21 | 27.64 | 35.8 | 66.9 | 42.2 | 51.9 | 60.4 | 32.5 | 39.5 | 60.4 | 48.69 |
| DEEPTRANSFORMERS (ours) | 20.32 | 20.67 | 36.9 | 68.4 | 49.8 | 52.8 | 65.7 | 34.9 | 40.2 | 61.8 | 51.31 |
| DOT (ours) | 19.96 | 20.15 | 39.0 | 69.1 | 50.7 | 53.1 | 66.2 | 37.0 | 40.3 | 63.7 | 52.39 |
| RetNet | 26.08 | 24.45 | 34.5 | 67.2 | 41.6 | 52.1 | 63.2 | 32.8 | 38.4 | 57.9 | 48.46 |
| DeltaNet | 24.37 | 24.60 | 37.1 | 66.9 | 42.0 | 50.7 | 64.9 | 31.4 | 39.9 | 59.0 | 48.97 |
| TTT | 24.17 | 23.51 | 34.7 | 67.3 | 43.9 | 51.0 | 64.5 | 33.8 | 40.2 | 59.6 | 47.32 |
| Gated DeltaNet | 21.18 | 22.09 | 35.5 | 68.0 | 44.9 | 50.7 | 66.9 | 33.1 | 39.2 | 59.1 | 49.69 |
| Samba* | 20.63 | 22.71 | 39.7 | 69.2 | 47.4 | 52.0 | 66.9 | 33.2 | 39.0 | 61.2 | 51.08 |
| Gated DeltaNet-H2* | 19.88 | 20.83 | 39.2 | 69.0 | 48.2 | 52.6 | 67.0 | 35.5 | 39.4 | 61.1 | 51.49 |
| Titans (LMM) | 20.04 | 21.96 | 37.4 | 69.3 | 48.5 | 52.3 | 66.3 | 35.8 | 40.1 | 62.8 | 51.56 |
| MONETA | 21.18 | 21.94 | 38.02 | 69.55 | 49.16 | 53.01 | 67.47 | 36.09 | 40.53 | 63.18 | 52.12 |
| MEMORA | 22.28 | 22.31 | 38.2 | 67.8 | 49.3 | 53.3 | 63.6 | 36.1 | 40.9 | 63.0 | 51.52 |
| SWDT (ours) | 19.89 | 21.52 | 36.2 | 68.3 | 45.2 | 53.0 | 65.4 | 34.2 | 39.5 | 59.5 | 50.1 |
| DLA (ours) | 23.12 | 22.09 | 36.1 | 68.0 | 47.9 | 52.7 | 65.8 | 34.6 | 39.1 | 59.6 | 50.46 |
| OMEGANET (ours) | 19.16 | 20.14 | 38.7 | 69.8 | 50.0 | 53.3 | 67.8 | 36.8 | 39.6 | 64.4 | 52.56 |
| ATLAS (ours) | 18.92 | 21.01 | 39.1 | 69.7 | 50.2 | 53.5 | 67.5 | 37.1 | 40.7 | 64.3 | 52.77 |
| ATLAS++ (ours) | 19.04 | 20.03 | 39.7 | 69.7 | 51.1 | 53.2 | 68.2 | 37.4 | 40.9 | 64.4 | 53.09 |
| ATLAS (MAG) | 18.62 | 21.18 | 40.0 | 70.3 | 50.5 | 53.0 | 68.1 | 36.5 | 41.2 | 65.0 | 53.08 |
| ATLAS (MAL) | 19.07 | 21.46 | 38.8 | 69.2 | 50.5 | 53.6 | 67.3 | 36.1 | 41.0 | 64.5 | 52.63 |
| | | | | 1.3B params / 100B tokens | | | | | | | |
| Transformer++ | 18.53 | 18.32 | 42.6 | 70.0 | 50.2 | 53.5 | 68.8 | 35.1 | 40.7 | 57.1 | 52.25 |
| DEEPTRANSFORMERS (ours) | 15.67 | 12.63 | 49.4 | 72.6 | 57.0 | 58.8 | 71.1 | 37.5 | 41.6 | 61.5 | 56.19 |
| DOT (ours) | 15.28 | 11.96 | 50.1 | 73.3 | 57.5 | 60.4 | 72.2 | 41.2 | 42.7 | 61.4 | 57.35 |
| RetNet | 19.08 | 17.27 | 40.5 | 70.1 | 49.2 | 54.1 | 67.3 | 33.8 | 40.8 | 60.4 | 52.02 |
| Mamba2 | 16.56 | 12.56 | 45.7 | 71.9 | 55.7 | 55.2 | 72.5 | 37.9 | 40.2 | 60.1 | 54.89 |
| DeltaNet | 17.71 | 16.88 | 42.5 | 70.7 | 50.9 | 53.3 | 68.5 | 35.7 | 40.2 | 55.3 | 52.14 |
| Gated DeltaNet | 16.42 | 12.17 | 46.6 | 72.2 | 55.8 | 57.4 | 71.2 | 38.4 | 40.6 | 60.2 | 55.32 |
| Samba* | 16.13 | 13.29 | 44.9 | 70.9 | 53.4 | 55.6 | 68.8 | 36.2 | 40.0 | 62.1 | 54.00 |
| Gated DeltaNet-H2* | 15.91 | 12.55 | 48.8 | 72.2 | 56.9 | 57.8 | 71.4 | 39.1 | 41.2 | 61.6 | 56.18 |
| MONETA | 15.52 | 11.47 | 47.88 | 73.16 | 56.14 | 59.09 | 72.53 | 40.32 | 41.91 | 61.18 | 56.52 |
| YAAD | 15.18 | 11.89 | 47.23 | 72.81 | 56.46 | 59.02 | 72.14 | 40.05 | 40.73 | 61.86 | 56.39 |
| Titans (LMM) | 15.60 | 11.41 | 49.1 | 73.1 | 56.3 | 59.8 | 72.4 | 40.8 | 42.1 | 61.0 | 56.82 |
| MEMORA | 15.90 | 12.04 | 48.7 | 73.1 | 56.0 | 57.4 | 71.5 | 37.9 | 40.2 | 61.3 | 55.87 |
| OMEGANET (ours) | 14.91 | 11.26 | 49.7 | 73.4 | 57.6 | 59.7 | 72.6 | 40.3 | 42.4 | 62.1 | 57.23 |
| ATLAS (ours) | 14.97 | 10.98 | 50.1 | 73.9 | 57.3 | 60.2 | 72.8 | 41.0 | 42.9 | 62.8 | 57.62 |
| ATLAS++ (ours) | 14.40 | 10.72 | 50.8 | 73.5 | 59.4 | 61.1 | 71.3 | 43.7 | 42.5 | 61.9 | 58.03 |

Comparing Transformer++ with our more generalized Transformers (i.e., DEEPTRANSFORMERs, and DOT) we observe a consistent performance improvement. We attribute this performance to their deep memory, which makes them more powerful to model the dependencies of tokens. Comparing DOT with DEEPTRANSFORMERs, we can see the advantage of Omega rule, which helps the model to better manage its memory.

## J.2 LEARNABILITY EXPERIMENTS

We have also performed some small-scale experiments to analyze the function-learning capability of small MLPs in an online fashion. In this setting, we have a sequence of tuples $(i_1, o_1), \ldots (i_t, o_t)$ with both $i_j, o_j \in \mathbb{R}^d$ for all $j$. We train an MLP $\mathcal{M}$ in an online fashion to minimize $\text{loss}_j = \|i_j - o_j\|_2^2 / \|o_j\|_2^2$ – specifically, we compute the gradient at time step $j$ as $\nabla_{\mathcal{M}.\text{params}} \text{loss}_j$ and use standard optimizers such as Adam, Rmsprop and SGD to update the parameters. Such experiments

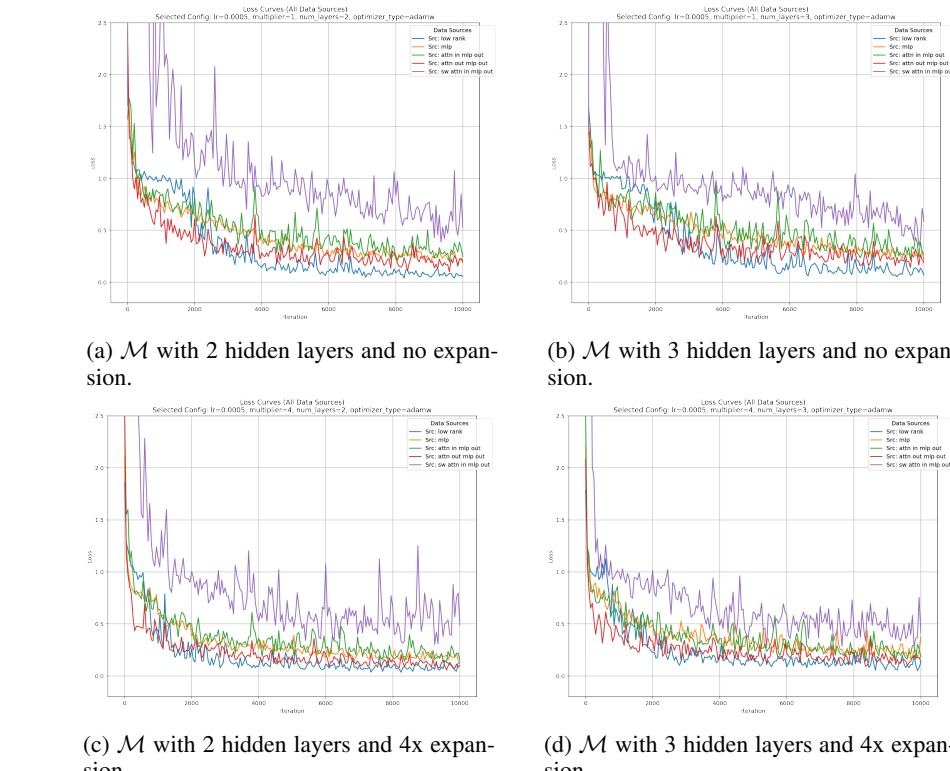

(a) $\mathcal{M}$ with 2 hidden layers and no expansion.

(b) $\mathcal{M}$ with 3 hidden layers and no expansion.

(c) $\mathcal{M}$ with 2 hidden layers and 4x expansion.

(d) $\mathcal{M}$ with 3 hidden layers and 4x expansion.

Figure 5: Loss curves for different setting with various hyperparameters

help us understand the representation power of the models we use to represent memory and the power of optimization algorithms to quickly learn the underlying sequence mapping.

We study five different sequence to sequence functions:

1. **Low Rank Mappings**: We sample a random low rank matrix $\mathbf{W} = \mathbf{XY}$ with $\mathbf{X} \in \mathbb{R}^{d \times k}$ and $\mathbf{Y} \in \mathbb{R}^{k \times d}$. We then sample $i_1, \ldots, i_t$ randomly from a Gaussian distribution and set $o_j = \mathbf{W}^\top \cdot i_j$ for all $j \in [t]$.

2. **MLP Mappings**: We sample an MLP $\mathcal{M}$ with 1 input, 1 hidden and 1 output layer which uses GELU non-linearity. We set the hidden dimension to $d$ so that there is no expansion. We then sample $i_1, \ldots, i_t$ randomly from a Gaussian distribution and then set $o_j = \mathcal{M}(i_j)$ for all $j \in [t]$.

3. **Attention+MLP Mapping**: We sample $(i_1, \ldots, i_t)$ from a Gaussian distribution and an MLP $\mathcal{M}$ as above. We additionally sample three $d \times d$ matrices $\mathbf{W_Q}$, $\mathbf{W_K}$ and $\mathbf{W_V}$ and compute $q_j = \mathbf{W_Q}^\top \cdot i_j$, $k_j = \mathbf{W_K}^\top \cdot i_j$ and $v_j = \mathbf{W_K}^\top \cdot i_j$ for all $j \in [t]$. We then compute $o_1', \ldots, o_t'$ as outputs of the causal masked attention mechanism applied on $\{q_j\}_{j \in [t]}, \{k_j\}_{j \in [t]}, \{v_j\}_{j \in [t]}$ and finally compute $o_j = \mathcal{M}(o_j)$.

4. **Attention Outputs as Inputs**: We do the same as above except that we output $o_j'$ as the input sequence and $o_j$ as the output sequence.

5. **Sliding Window Attention + MLP Mapping**: We do the same as in **Attention + MLP Mapping** setting except that we use a sliding window attention instead of full attention. We use a sliding window of 512 in our experiments.

Note that the settings 3 and 5 are much harder to learn since they require (partially) memorizing the previous inputs and outputs to be able to learn the function that maps $i_j$ to $o_j$, whereas the settings 1, 2 and 4 do not need to memorize the previous input-output pairs and just need to learn the underlying low-rank matrix or the MLP that maps the inputs to outputs.

The setting 4 is slightly different to setting 2 in that the inputs are not-independent at each time step and are correlated by the attention mechanism we use to compute the inputs. Thus a strong learning algorithm maybe able to utilize the underlying correlations to learn the mapping faster in setting 4 versus setting 2.

We set $d = 256$ and show the loss curves vs sequence position for all the five settings with function learning MLP $\mathcal{M}$ being defined and trained with different settings in Figure 5. We can see that in all the settings, the model learns non-trivial mappings from inputs to outputs with the $loss_j = \|i_j - o_j\|_2^2 / \|o_j\|_2^2$ being smaller than 1 eventually. Most notably, the correlations in inputs in setting 4 induced by the attention mechanism makes the model quickly learn the mapping compared to in setting 2 and the models usually learn the best in setting 1 which is the least complex function.

The models do the worst in settings 3 and 5 which require the models to (partially) memorize the inputs and outputs to learn the attention mechanism outputs. Surprisingly, the models learn to do better in setting 3 vs setting 5, when we would expect that capacity requirement for setting 3 to be higher than setting 5. We hypothesize that the learning algorithm is unable to make the model 'forget' old inputs which makes the loss worse in sliding window setting when compared to global attention setting. A caveat of our analysis is that, the attention computation is done on randomly initialized vectors and hence the attention matrix is usually not spiky, unlike in the attention matrix for trained set of query, key and value vectors in LLMs. This leads to attention outputs being close to the mean of value vectors in the context.

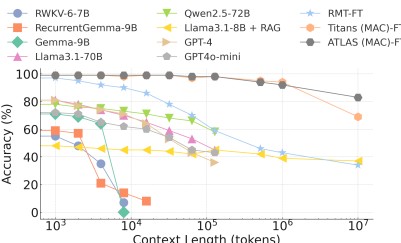 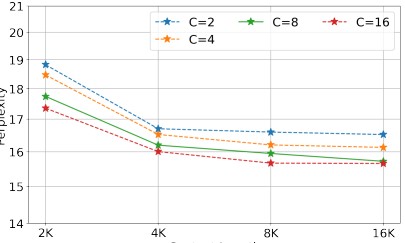

Figure 6: Performance of ATLAS and baselines on BABILong benchmark. ATLAS surpasses Titans performance and effectively scale to 10M context length in this task.

Figure 7: The effect of local context length (i.e. $c$) on the performance of OMEGANET with different global context length.

### J.3 LONG CONTEXT: BABILONG BENCHMARK

To compare the effectiveness of ATLAS with Titans (Behrouz et al., 2024) in ultra-large sequences, we further evaluate ATLAS's performance on BABILong benchmark (Kuratov et al., 2024). In this experiment, we follow Behrouz et al. (2024) and use MAC architecture but without persistent memory tokens. We also follow the original setup in the benchmark and fine-tune our model. The results are reported in Figure 6. While ATLAS shows competitive and on par performance with Titans until 1M context length, the performance of Titans drops in 10M. ATLAS, however, maintains its performance and achieve +80% accuracy in 10M context length. We attribute this to more powerful memory; in terms of (1) memory management (i.e., the use of Muon), (2) better memory capacity due to polynomial kernels, and (3) its nature to memorize the context, instead of individual tokens.

In previous sections, we show the effectiveness of our Transformer-like architectures (i.e., DEEP-TRANSFORMERs and DOT) in both language modeling and long-context needle-in-haystack tasks. From now on, we focus on our recurrent architectures (i.e., ATLAS, and OMEGANET) to show the importance of presented improvements.

Table 6: Performance of ATLAS, OMEGANET, and baselines on the synthetic benchmark of MAD (Poli et al., 2024). ATLAS outperforms all the baselines, including Transformers.

| | Compression | (Noisy) ICR | Fuzzy ICR | Selective Copying | Memorization | Average |
|---|---|---|---|---|---|---|
| Transformers | 49.4 | 100 | 48.2 | 95.9 | 83.8 | 75.46 |
| Gated DeltaNet | 44.8 | 100 | 32.5 | 96.2 | 81.7 | 71.04 |
| Titans | 49.6 | 100 | 49.7 | 99.4 | 83.5 | 76.44 |
| OMEGANET (ours) | 50.9 | 100 | 54.2 | 99.6 | 90.2 | 78.98 |
| ATLAS (ours) | 51.6 | 100 | 54.9 | 99.6 | 91.4 | 79.50 |

Table 7: The performance of our models (ATLAS, and OMEGANET) compared to baselines. While still Transformers achieve the best results in in-context recall tasks, our design of context memorization and polynomial feature maps can close the gap with Transformers.

| | SWDE | NQ | DROP | FDA | SQUAD | TQA | Average |
|---|---|---|---|---|---|---|---|
| Transformers | 84.9 | 23.0 | 28.4 | 72.5 | 48.1 | 64.4 | 53.55 |
| Gated DeltaNet | 63.2 | 19.1 | 26.7 | 33.4 | 39.6 | 59.7 | 40.28 |
| Titans | 65.1 | 20.7 | 27.2 | 37.3 | 42.6 | 61.0 | 42.31 |
| OMEGANET (ours) | 67.4 | 21.1 | 27.2 | 39.0 | 43.2 | 60.9 | 43.13 |
| ATLAS (ours) | 66.8 | 21.9 | 27.4 | 40.7 | 44.1 | 61.3 | 43.70 |

Table 8: Ablation Study on ATLAS. All components of ATLAS are positively contributing to its performance.

| Model | Language Modeling ppl ↓ | C.S. Reasoning acc ↑ |
|---|---|---|
| ATLAS | 19.97 | 52.77 |
| +Gated MLP Memory | 19.53 | 53.09 |
| +Attn (MAG) | 19.90 | 53.08 |
| +Attn (MAL) | 20.26 | 52.63 |
| Linear Memory | 21.03 | 49.74 |
| w/o Muon | 19.65 | 52.56 |
| c = 1 | 21.98 | 49.26 |
| w/o Polynomial Mapping | 22.14 | 50.57 |

## J.4 ADDITIONAL EXPERIMENTS: IN-CONTEXT RECALL, MAD SYNTHETIC BENCHMARK, AND ASSOCIATIVE RECALL

In this section, we first evaluate the performance of our models on MAD benchmark, a synthetic benchmark that evaluate the performance of models in recall, memorization, compression, and copying tasks (Poli et al., 2024). The results are reported in Table 6. ATLAS achieves the best results in all aspects, particularly in memorization, which shows the importance of its components for enhancing the memory capacity.

In-context recall tasks is one of the most challenging benchmarks for recurrent neural networks. In this section, we follow Arora et al. (2024) and perform experiments on SWDE (Lockard et al., 2019), NQ (Kwiatkowski et al., 2019), DROP (Dua et al., 2019), FDA (Arora et al., 2023b), SQUAD (Rajpurkar et al., 2016), and TQA (Kembhavi et al., 2017) to evaluate and compare the performance of ATLAS with baselines and Transformers. The results are reported in Table 7. While Transformers still achieve the best results in in-context recall tasks, ATLAS and OMEGANET shows competitive performance and performs better than state-of-the-art recurrent models. We again attribute this performance to better memory management and capacity.

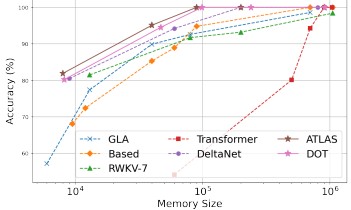 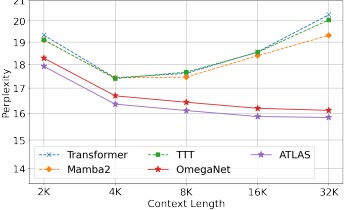 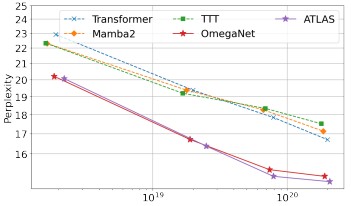

Figure 8: The results for associative memory recall.

Figure 9: Scaling patterns of ATLAS, and OMEGANET with respect to (Left) training context length, and (Right) FLOPs.

Finally, following Yang et al. (2024c) and Arora et al. (2023a) we evaluate the performance of ATLAS and DOT in Multi-Query Associative Recall (MQAR) task (Arora et al., 2023a). The results are reported in Figure 8. Both models show good performance compared to baselines and ATLAS achieve the best performance per memory size compared to state-of-the-art models such as DeltaNet (Yang et al., 2024c).

## J.5 ABLATION STUDY AND SCALING PATTERNS

In this section, we perform an ablation study on the differernt components of ATLAS, and also evaluate its scaling patterns with respect to the number of parameters and also the context length of the training. The results for ablation study are reported in Figure 2. The results show that: (1) more powerful memory architectures such as gated MLP can further enhance the performance of ATLAS; (2) The hybrid variants further improve the performance, where MAG shows better improvement compared to MAL architecture; (3) Polynomial mappings as well as deep memory are particularly important when we use context memorization (i.e., Omega rule). Figure 7 also shows the effect of local context length (i.e., $c$) on the performance of the model. With the increase of $c$ we can achieve better performance, mainly due to the gating parameters of $\gamma$ that can prune the context, whenever it is needed.

**Model Size.** Figure 9 shows the scaling pattern of ATLAS, and OMEGANET, with respect to number of parameters and compared to baseline. Both models achieve a good scaling pattern with increasing the model size, achieving lower perplexity in all scales compared to baselines.

**Context Length.** Figure 9 shows the scaling pattern of ATLAS, and OMEGANET, with respect to the context length and compared to baseline. Both models due to high memory capacity can scale well, when increasing the context length.

**The Effect of $p$.** We first, evaluate the effect of $p$ on the training throughput and perplexity of the model. We consider the context length of 2K, 4K, 8K. The results are in Figures 10 and 11 .

We further evaluate the memory usage. If we consider the case $p = 1$ as the base, $p = 2$, $p = 3$, and $p = 4$ requires $\times 1.2$, $\times 2.5$, and $\times 3.3$, respectively.

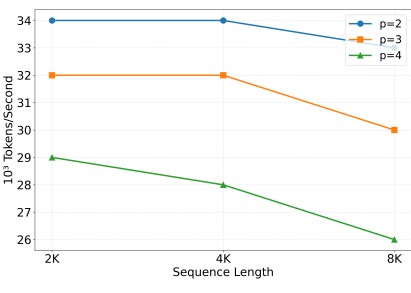

Figure 10                          Figure 11

