# OpenReview forum: "ATLAS: Learning to Optimally Memorize the Context at Test Time"
_ICLR.cc/2026/Conference — Submitted to ICLR 2026_

### Official Review · Reviewer_Fg6k · 2025-10-26

**Soundness:** 3
**Presentation:** 3
**Contribution:** 3
**Rating:** 6
**Confidence:** 2

**Summary:**

The paper investigates into ways to improve recurrent architectures. More precisely, for recurrent architectures, the paper investigates into the memory modelling capacity, the paper made the following contributions in this area:
- theoretical understanding of memory capacity and a novel way to increase this capacity via kernels (polynomial kernels implemented)
- using sliding window updates instead of updates over single tokens for memory update
- adopt a more suitable optimizer (Muon) for the proposed architectures

Experimental results show that the proposed changes translate to general task improvements as well as to the long context task, measured via some RULER benchmarks. The proposed changes are further ablated to show their effectiveness.

**Strengths:**

The paper has made contributions with a focus of long term tasks for recurrent architecture memories. The investigation into expressivity seems to be the main novelty of the paper since it contains a novel technical solution (polynomial kernel) with solid theoretical motivations. The sliding window idea has already been implemented in architectures such as MAG and MAL but the paper makes further steps to integrate the idea into a single architectures.

All the contributions have been tested not only in long context tasks (which is the focus of the paper in motivations) but also on other popular benchmarks to give an overview of the model performance. The proposed components and contributions have been mostly properly ablated.

**Weaknesses:**

From Table 2, Muon does not offer great improvement and it seems this is not a central contribution from the paper. If my understanding is correct, it seems this part can be removed from the main section of the paper.

Maybe this is more of a question, why Atlas++ results are not shown in Table 2?

**Questions:**

MAG and MAL architectures contain sliding window improvement (in spirit similar to Omega rule proposed in this paper), so:
- Why MAG and MAL gives further improvement over Atlas which already has this sliding window aspect included in its design (i.e. Table 1 comparinsg MAG/MAL to Atlas)
- Is there any computational advantages that the current paper can claim over MAG/MAL which are hybrid models? It seems to me the answer is positive.

---

> ### Author Response · Authors · 2025-12-04
>
> Thank you for your time and detailed feedback. We answer your questions below.
>
> > From Table 2, Muon does not offer great improvement and it seems this is not a central contribution from the paper. If my understanding is correct, it seems this part can be removed from the main section of the paper.
>
> **Response:**
> Please note that for all design choices we have provided either theoretical and/or emperical evidences. For the use of Muon, we have performed ablation studies and showed that removing Muon and using simple gradient descent can damage the performance. Please note that the performance gain of having Muon is relatively significant for the common-sense reasoning tasks. For example, we kindly bring to your consideration that big transitions such as transition from Mamba2 to Gated DeltaNet (i.e., changing the entire learning rule from Hebbian rule to Delta rule) has provided +0.4 performance gain while in our case, simply the use of Muon (comparison of Atlas with OmegaNet in Table 1) has provided +0.4 performance gain in 1.3B scale.

---

### Official Review · Reviewer_KJpp · 2025-11-01

**Soundness:** 2
**Presentation:** 3
**Contribution:** 3
**Rating:** 6
**Confidence:** 4

**Summary:**

The paper introduces Atlas, a new long-term memory module for sequence models based on test-time optimization of memory states. Atlas tries to fix three common limits of modern recurrent networks. First, limited capacity of memory states by using polynomial kernels. Second, it writes local updates to memory with a context-aware sliding window called Omega, recurrent update rule based not only the current token. Third, more expressive memory update rule with updates from Muon optimizer instead of GD. The authors also propose a family of related models (DLA, SWLA, Omega, Atlas) and compare them with previous works such as Titans and modern recurrent models.

**Strengths:**

- Nice idea and clean motivation. The paper clearly explains the problems with memory state capacity, state update locality, and state optimization and how Atlas addresses them.
- Systematization of prior work. It provides a helpful, unified perspective on earlier memory modules through attentional bias and test-time optimization (particularly Table 3)
- Results on language modeling, language understanding and needle-in-a-haystack tasks show strong performance. Ablation study supports design choices.

**Weaknesses:**

- Impact of polynomial mapping (value of p) on benchmarks results is not clear, especially on long context ones and MAD tasks, where larger memory capacity should show its benefit.
- Features claimed for Omega (sliding window, global context-aware memory updates) are already in Block-Recurrent transformers, RMT, ARMT, and MELODI, which use the Transformer as a recurrent cell. Therefore, the novelty in this case is unclear. The text still lacks this discussion.
- On BABILong, results are shown (Fig. 6) but ARMT is missing. The BABILong paper (Figure 1, Table 2) reports ARMT as the top model (~77 average accuracy on QA1-5 at 10M tokens) and shows its scaling to 50M tokens. Please include and discuss ARMT for completeness.
- Evaluations on long-contexts, which are one of the key stated features of Atlas model, are only up to 16k tokens on the RULER benchmark. Currently, 16k tokens is not a very long context. More impressive Atlas results are shown on the BABILong benchmark in the appendix, with strong performance on up to 10 million tokens.
- No results are reported with standard deviations, and the authors do not indicate how many runs were performed. Many reported numbers are close to those of other methods.

**Questions:**

- Please address limitations as questions.
- what exact value of p (polynomial mapping) is used in experiments, it is not mentioned in the paper.
- several Titans variants exist (MAC, MAG, MAL); why was a weaker one used? Could you add discussion and comparison with the strongest published versions. Add Atlas MAC, MAG, MAL vs Titans MAC, MAG, MAL comparison.
- SWDT, what is it? Table 1, Table 5. Where are the results of SWLA?
- Are reported scores zero-shot or after fine-tuning (e.g., Table 1, 2)? Please specify for each task.


typos:
- L210, l^(1) misses non-linearity \phi applied to key ?
- Fig. 11, y label should be perplexity?

---

> ### Author Response · Authors · 2025-12-04
>
> Thank you for your time and detailed feedback. We answer your questions below.
>
> > Impact of polynomial mapping (value of p) on benchmarks results is not clear, especially on long context ones and MAD tasks, where larger memory capacity should show its benefit.
>
>
>
> **Response:** Please note that in Figure 10 and 11 in Appendix, we study the effect of value of $p$ on the performance and efficiency of the model. Please note that considering its effect for tasks such as MAD is challenging, mainly due to the fact that using polynomial mapping we are also increasing the state size for these models and so we cannot evaluate the performance of the model when we control the state size.
>
>
>
>
> > Features claimed for Omega (sliding window, global context-aware memory updates) are already in Block-Recurrent transformers, RMT, ARMT, and MELODI, which use the Transformer as a recurrent cell. Therefore, the novelty in this case is unclear. The text still lacks this discussion.
>
>
> **Response:**  Thank you for your suggestion. We will add a detiled discussion about the differences with block-recurrent transformers. Please note that our architecture is still attention free and does not have any form of Transformers or attention. On the other hand, block-recurrent transformers are based on softmax global attention and use the state of the transformer for the recurrence.
>
>
>
>
>
> > On BABILong, results are shown (Fig. 6) but ARMT is missing. The BABILong paper (Figure 1, Table 2) reports ARMT as the top model (~77 average accuracy on QA1-5 at 10M tokens) and shows its scaling to 50M tokens. Please include and discuss ARMT for completeness.
>
>
> **Response:** Thank you for your suggestion. Following your suggestion, we have added a discussion about ARMT and its results for the BABILong benchmark. Please note that ARMT and Atlas can potentially be combined and so are orthogonal. That is, ARMT is based on delta-rule and one can change it to Omega rule to gain even better performance.
>
>
>
>
> > Evaluations on long-contexts, which are one of the key stated features of Atlas model, are only up to 16k tokens on the RULER benchmark. Currently, 16k tokens is not a very long context. More impressive Atlas results are shown on the BABILong benchmark in the appendix, with strong performance on up to 10 million tokens.
>
>
>
> **Response:** Thank you, following your suggestion, in the final version, we bring the result of BABILong benchmark to the main part of the paper.
>
> Also, please note that even in 16K, the baseline recurent models are showing around zero performance. Going beyond that for RULER tasks cannot help us to show the superiority of our design.
>
>
>
> > No results are reported with standard deviations, and the authors do not indicate how many runs were performed. Many reported numbers are close to those of other methods.
>
> **Response:** Please note that running these experiments multiple times can be extremly costly and might not be possible for all researchers to afford the cost. Also, we have followed other peer studies, and similar to them, only use 1 run, which can also help with cost management.
>
>
>
>
> > what exact value of p (polynomial mapping) is used in experiments, it is not mentioned in the paper.
>
> **Response:** Thank you, we use $p=2$. We will add this to the final version of the paper.
>
>
> > Several Titans variants exist (MAC, MAG, MAL); why was a weaker one used? Could you add discussion and comparison with the strongest published versions. Add Atlas MAC, MAG, MAL vs Titans MAC, MAG, MAL comparison.
>
> **Response:** Please note that we have used both MAG and MAL variants and only MAC architecture is missing. The main reason, as also has been discussed by Titans paper, is MAC architecture provides a trade-off between effieicny and effectiveness and so a fair comparison is extremely challenging.

---

### Official Review · Reviewer_Bhve · 2025-11-01

**Soundness:** 2
**Presentation:** 3
**Contribution:** 3
**Rating:** 4
**Confidence:** 4

**Summary:**

The paper proposes ATLAS, a recurrent long-term neural memory module that (i) increases capacity via higher-order (polynomial) key/query features, (ii) replaces online token-wise updates with an Omega sliding-window rule intended to memorize the context, instead of tokens, and (iii) improves memory management by using a Muon/second-order–like inner optimizer. Models (OMEGANET, ATLAS, and hybrid MAG/MAL variants) are evaluated on language modeling, common-sense reasoning, RULER NIAH, and several recall tasks.

**Strengths:**

1. Theory is interesting and clearly framed. The capacity analysis (matrix vs deep memory; effect of polynomial mappings) and the Omega sliding-window objective are neat, well-motivated contributions.

2. Broad empirical sweep with solid baselines from modern RNN families; results on RULER NIAH and standard LM/CS tasks are competitive, often best among non-hybrids.

**Weaknesses:**

1. Key claims are not directly stress-tested. The headline contribution - “a long-term neural memory module with high capacity and the ability to memorize the context, instead of tokens" - is asserted, but experiments do not isolate these mechanisms. Per-benchmark wins and one ablation (showing c=1 hurts) are helpful, yet there is no targeted evaluation that would uniquely validate context-level memorization vs token-wise memorization (e.g., tasks constructed so token-level memorization provably fails while context-level succeeds), nor a controlled capacity–vs–memory-size curve demonstrating the super-linear capacity predicted by the theory. The paper cites MAD/MQAR/recall results in the appendix, but these still read as broad benchmarks rather than diagnostic tests tied to the two claims above; the manuscript does not analyze why Omega/polynomial/Muon are winning on those axes.

2. Baseline gap for Transformers. Table 1 uses an internal Transformer++ control, but omits widely used, publicly available ~1B open-weight Transformers (e.g., LLaMA-/Qwen) trained on comparable token budgets. If ATLAS is posited as a RNN alternative for Transformers, it should be compared head-to-head with similarly sized, well-known Transformer baselines under matched data/compute.

3. Sparse discussion/analysis. The results sections are super brief and largely descriptive (“we attribute this to memorizing context / polynomial kernels”). There is little error analysis (where ATLAS fails vs Titans/DeltaNet/Transformers), no probing of Omega gates γ, and limited introspection on the Muon inner optimizer beyond citing second-order intuition.

4. Muon ablation paradox (Fig. 2). Muon is introduced to avoid poor local optima that hurt long-context performance, yet removing Muon improves perplexity on the LM task in Fig. 2. This apparent contradiction is neither analyzed nor discussed.

5. Hybrids (MAL/MAG) lack description in this paper. Scores are reported for MAL and MAG hybrids of ATLAS, but the paper does not explain what MAL/MAG are or why they matter beyond citing prior work. Including hybrid results without even a brief motivation and description makes them difficult to interpret and dilutes the empirical story. Either describe them succinctly (what they add, how they interact with ATLAS) or move them to appendix with clearer rationale.

6. Related work coverage. Given the positioning, I expected a clearer discussion of Transformer variants with segment-level recurrence/sliding-window attention as alternative ways to achieve context-level optimization (e.g., SWA/Recurrent Memory Transformer/Associative Recurrent Memory Transformer and others). Segment-recurrent transformers have memory for context and thus directly related to one of the core contributions of the paper - token vs. context memorisation. The paper notes the connection in passing but does not compare empirically to such Transformer baselines.

**Questions:**

1. Can you design diagnostic tasks that decisively require context-level memorization (not token-wise), and report ATLAS vs c=1 and token-memorizing RNNs on them?

2. Please provide a capacity study: number of independent (key,value) pairs recovered vs memory size/degree-p features, with and without Muon. This would directly test the “high capacity” claim.

3. Why are LLaMA/Qwen-scale Transformer baselines absent for ~1B models? Could you add matched-compute comparisons (same tokens/context window) to strengthen the “Transformer alternative” narrative?

4. What are MAL/MAG, why include them, how do they interact with Omega/polynomial features?

5. Why does removing Muon yield better LM perplexity in Fig. 2 if Muon is meant to avoid bad local optima in long contexts?

---

> ### Author Response · Authors · 2025-12-04
>
> Thank you for your time and detailed feedback. We answer your questions below.
>
> > Key claims are not directly stress-tested.
>
> **Response:** Thank you, we want to kindly bring to your consideration that this is a 10-page research paper and defentely there are a lot of interesting directions for future work to explore and better understand the current designs. In this work, however, we have tried to provide **enough** emperical and theoretical support for our design choices. Particularly, in our ablation study, we have shown that removing Omega rule (i.e., using c=1) can damage the performance significantly. In fact, it has been the most effect component and its removal has caused the largest performance drop in both language modeling and common-sense reasoning tasks.
>
> In Figure 7, we specifically study the effect of $c$ on the performance and we ablate its value. We believe that all these improvemens and performance gain are supporting the importance of Omega rule.
>
>
>
> > Baseline gap for Transformers. Table 1 uses an internal Transformer++ control, but omits widely used, publicly available ~1B open-weight Transformers (e.g., LLaMA-/Qwen) trained on comparable token budgets.
>
>
> **Response:** Please note that when want to fairly compare sequence model backbones, it is important to use the same training process, making sure that all other compoenents are controled. Therefore, we use the baseline that are trained in the same envirenoment. Pre-trained models like Qwen or Llama use different datasets, different tokenization process and a lot of other design choices are different. Therefore, similar to most peer studies, we have not compared with pre-trained models as the comparison is not fair and also components are not controled so we could fully evaluate the effect of our designs.
>
>
>
> > Sparse discussion/analysis. The results sections are super brief and largely descriptive (“we attribute this to memorizing context / polynomial kernels”). There is little error analysis (where ATLAS fails vs Titans/DeltaNet/Transformers), no probing of Omega gates γ, and limited introspection on the Muon inner optimizer beyond citing second-order intuition.
>
> **Response:** Please note that for all design choices we have provided either theoretical and/or emperical evidences. For the use of Muon, we have performed ablation studies and showed that removing Muon and using simple gradient descent can damage the performance. Please note that the performance gain of having Muon is relatively significant for the common-sense reasoning tasks. For example, we kindly bring to your consideration that big transitions such as transition from Mamba2 to Gated DeltaNet (i.e., changing the entire learning rule from Hebbian rule to Delta rule) has provided +0.4 performance gain while in our case, simply the use of Muon (comparison of Atlas with OmegaNet in Table 1) has provided +0.4 performance gain in 1.3B scale.
>
> Similarly, please note that OmegaNet is the superset of DeltaNet and when we use $c=1$, the Omega learning rule collapses into Delta rule. Following our ablations, we have shown that the case of $c=1$ can damage the performance. The similar point applies to Titans as well. In fact, using gradient descent with momentum and orthogonalization in Muon are generalized case of Titans that use simple gradient descent with momentum and with Delta update rule.
>
> Similarly, for the gating of $\gamma$, removing the gating results in collapsing the design into online learner. However, following your suggestion, we have added a new row to the ablation study and remove $\gamma$. The results are 21.6 and 50.3 for perplexity and common-sense reasoning, which means that the performance drops signifcantly if we remove these components.
>
>
> > Muon ablation paradox (Fig. 2). Muon is introduced to avoid poor local optima that hurt long-context performance, yet removing Muon improves perplexity on the LM task in Fig. 2. This apparent contradiction is neither analyzed nor discussed.
>
> **Response:** Please note that when we scale the model, the performance of with and without Muon optimizer can be observed. For example comparing the language modeling results for 1.3B model for OmegaNet (w/o muon) and Atlas (with muon) shows performance gain even in perplexity.
>
> Please note that even without Muon, we have OmegaNet, SWLA, DLA, etc.,  all showing better performance than state-of-the-art methods and also all have novel elements such as new learning rules, novel combinations with deep memory, etc. Therefore, we believe that our contributions have provided enough new technical results and also supported them with emperical and theoretical evaluations.

---

> > ### Author Response · Authors · 2025-12-04
> >
> > > Hybrids (MAL/MAG) lack description in this paper. Scores are reported for MAL and MAG hybrids of ATLAS, but the paper does not explain what MAL/MAG are or why they matter beyond citing prior work. Including hybrid results without even a brief motivation and description makes them difficult to interpret and dilutes the empirical story. Either describe them succinctly (what they add, how they interact with ATLAS) or move them to appendix with clearer rationale.
> >
> > **Response:** Thank you, following your suggestion, we provide these descriptions from the prior works in the paper to make it more easy to follow. Both MAL and MAG are hybrid models, where we combine attention and memory module in a layer- or head-wise manner.
> >
> >
> >
> >
> > > Related work coverage. Given the positioning, I expected a clearer discussion of Transformer variants with segment-level recurrence/sliding-window attention as alternative ways to achieve context-level optimization (e.g., SWA/Recurrent Memory Transformer/Associative Recurrent Memory Transformer and others). Segment-recurrent transformers have memory for context and thus directly related to one of the core contributions of the paper - token vs. context memorisation. The paper notes the connection in passing but does not compare empirically to such Transformer baselines.
> >
> > **Response:** Please note that, as we also discussed in the related work, the segment-level recurrent models are orthogonal direction to ours and in fact, we can apply both of these techniqes together. In our evaluation, similar to all other studies in this area of desinging recurrent architectures, we have compared to fully recurrent sequence models backbones as well as Transformers. There are indeed a lot of interesting baselines to compare with, but it requires unlimited computing resources. Therefore, we have followed the literature in this sub-area and use the same types of baselines.

---

### Meta-Review · Area_Chair_EDyF · 2026-01-06

**Summary:**

This submission proposes ATLAS, a recurrent long-term memory module that aims to “memorize the context” at test time by addressing three limitations of existing memory models: 1, limited memory capacity via polynomial feature mappings, 2, local/online updates via an sliding-window objective, and 3, ess expressive memory management via a Muon (second-order–like) inner optimizer. The paper reports a broad empirical sweep across language modeling, commonsense reasoning, recall-intensive tasks, and long-context understanding and includes ablations for key components.

Overall, while the motivation is timely and the proposed components are interesting, the submission does not yet provide sufficiently diagnostic evidence to substantiate its core claims, and the empirical story is complicated—especially around Muon and hybrid results—leading to concerns of unclear attribution of gains.

Strengths

1. Clear, structured motivation: capacity, locality/online update limitations, and memory-state optimization are well-motivated.
2. Theoretical framing of capacity and the role of feature mappings is a useful for recurrent memory design.
3. Broad evaluation across several benchmark classes.

Key weaknesses
1. A consistent critique is that the paper asserts “context-level memorization” and “high capacity,” but provides mostly broad benchmark wins rather than targeted tests that isolate these phenomena. Reviewers request diagnostic tasks for this. Current ablations (e.g., c=1) are suggestive but not decisive; analysis of why Omega/polynomial/Muon helps on specific failure modes is limited.
2. Muon + hybrid story is complex. reviewers note mixed or even contradictory signals of Muon in experiments. Hybrid variants (MAG/MAL) improve results but were initially under-described, and it becomes unclear whether improvements come from ATLAS’s core memory design or from added attention.

**Reviewer Concerns:**

Addressed

Some missing details clarified
Commitments to expand related-work discussion

still outstanding
see weaknesses

**Reviewer Scores:**

Bhve likely remains ~4; core requests (diagnostics and Muon conflicts) not convincingly resolved.
KJpp: likely remains ~6; evaluation protocols remain concerns.
Fg6k: likely remains ~6; still have Muon/hybrid attribution concerns

---

### Decision · Program_Chairs · 2026-01-26

Reject